# Extensive disruption of protein interactions by genetic variants across the allele frequency spectrum in human populations

Robert Fragoza[1,2,8], Jishnu Das [3,4,8], Shayne D. Wierbowski[1,2], Jin Liang[1,2], Tina N. Tran[5,6], Siqi Liang [1,2], Juan F. Beltran[1,2], Christen A. Rivera-Erick[1,2], Kaixiong Ye[1], Ting-Yi Wang [1,2], Li Yao [1,2], Matthew Mort[7], Peter D. Stenson[7], David N. Cooper [7], Xiaomu Wei[1], Alon Keinan[1], John C. Schimenti[5], Andrew G. Clark[1,6] & Haiyuan Yu [1,2]

Each human genome carries tens of thousands of coding variants. The extent to which this variation is functional and the mechanisms by which they exert their influence remains largely unexplored. To address this gap, we leverage the ExAC database of 60,706 human exomes to investigate experimentally the impact of 2009 missense single nucleotide variants (SNVs) across 2185 protein-protein interactions, generating interaction profiles for 4797 SNV-interaction pairs, of which 421 SNVs segregate at > 1% allele frequency in human populations. We find that interaction-disruptive SNVs are prevalent at both rare and common allele frequencies. Furthermore, these results suggest that 10.5% of missense variants carried per individual are disruptive, a higher proportion than previously reported; this indicates that each individual's genetic makeup may be significantly more complex than expected. Finally, we demonstrate that candidate disease-associated mutations can be identified through shared interaction perturbations between variants of interest and known disease mutations.

[1] Department of Computational Biology, Cornell University, Ithaca, NY 14853, USA. [2] Weill Institute for Cell and Molecular Biology, Cornell University, Ithaca, NY 14853, USA. [3] Ragon Institute of MGH, MIT and Harvard, Cambridge, MA 02139, USA. [4] Department of Biological Engineering, Massachusetts Institute of Technology, Cambridge, MA 02139, USA. [5] Department of Biomedical Science, Cornell University, Ithaca, NY 14853, USA. [6] Department of Molecular Biology and Genetics, Cornell University, Ithaca, NY 14853, USA. [7] Institute of Medical Genetics, Cardiff University, Heath Park, Cardiff CF14 4XN, UK. [8] These authors contributed equally: Robert Fragoza, Jishnu Das. Correspondence and requests for materials should be addressed to H.Y. (email: haiyuan.yu@cornell.edu)

Recent explosive population growth has generated an excess of rare genetic variation segregating in human populations that likely plays a key role in the individual genetic burden of complex disease risk[1–6]. In agreement with this paradigm, large-scale whole-genome and whole-exome sequencing efforts have reported an excess of genetic variation in human genomes segregating at very low allele frequencies[3,4,6–9]. In particular, rare coding single nucleotide variants (SNVs) have been predicted to disproportionately impact protein function[4,7,10] in human genomes; however, methods and metrics for estimating the functionality of coding SNVs vary widely, and there is no consensus estimate for the number of functional variants per individual[11]. As such, a direct assessment of the functional impact of coding SNVs could prove indispensable to furthering our understanding on how segregating genetic variation influences complex traits and human disease.

Biological processes are likely regulated through intricate networks of protein and macromolecular interactions, as opposed to single proteins acting independently[12,13]. Researchers have accordingly identified a large number of mutations that disrupt these interactions; however, most of these perturbations correspond to synthetic mutations from scanning mutagenesis experiments[14–16], the vast majority of which do not occur naturally in human populations. For example, the SKEMPI database comprehensively collected the impact of 3047 mutations on protein-binding events published in the literature[17], only seven of which are listed as human population variants in ExAC[9]. Efforts to examine the impact of disease-associated mutations on protein function[18–20] are also limited because most of these mutations are very rare and consequently only impact a small number of individuals. The evolutionary context in which all genomic variants evolve is largely missing from such studies as a result.

In order to acquire a more representative understanding of the functional impact of human population variants on protein function, we leveraged the ExAC dataset of coding variants from 60,706 human exomes[9] to systematically evaluate the impact of 2009 missense SNVs, 811 of which are segregating at minor allele frequency (MAF) > 0.1% in human populations, across 2185 protein–protein interactions. We find that disruptive SNVs are strongly enriched at conserved protein loci and occur more prevalently at lower allele frequencies, underscoring the functional importance of disruptive variants uncovered by our assays. Moreover, we also determine that on average 10.5% of coding SNVs carried per individual are expected to impact protein–protein interactions, a rate much higher than indicated by previous reports[4,7,10]. Unexpectedly, while we observe an enrichment of functional SNVs at rare allele frequencies in agreement with previous literature[3,4,7,10], we also find that 9.6% of tested common variants with MAF > 10% perturbed protein interactions, indicating that many common variants are also functional. Finally, we show how candidate disease-associated mutations can be identified through matching interaction perturbation profiles with known disease-causing mutations. In this manner, we characterize a rare variant on the enzyme PSPH that significantly reduces its catalytic activity, as well as a rare variant on SEPT12 that results in male subfertility in CRISPR-edited mice, demonstrating the functional relevance of the interaction-disruptive SNVs reported here.

## Results

**Disruptive SNVs occur extensively across broad MAF ranges.** Alterations to protein–protein interactions can have deleterious consequences to fitness, particularly in human genetic disease[19,21,22] (Fig. 1a). As such, coding variation at interaction interfaces is mostly rare[23] and subject to evolutionary constraint[24,25]. In contrast, common variation is expected to be largely neutral and therefore unlikely to be extensively functional[26–28]. Nonetheless, notable exceptions exist, including APOE-epsilon 4, a risk-associated allele for Alzheimer's disease[29] (MAF = 18.4%), and the P12A polymorphism (MAF = 11.0%) of PPARG, which increases risk for type 2 diabetes[30]. Indeed, the extent to which MAF indicates whether an allele is disruptive to protein interactions remains largely unexplored. Hence, to systematically identify functionally relevant SNVs across rare to common allele frequencies, we constructed a resource of sequence-verified, single-colony clones for 2009 SNVs derived from three major databases: 1676 variants from ExAC[9], 204 Mendelian disease-associated mutations from HGMD[31], and 162 somatic mutations in cancer from COSMIC[32]. To avoid oversampling rare variants which dominate ExAC, we randomly selected alleles in ExAC across defined MAF bins ranging from singletons to very common alleles (Fig. 1b; see the section "Methods").

Upon constructing this resource, we then performed yeast two-hybrid (Y2H) experiments to measure the impact of these 2009 missense SNVs across 2185 human protein–protein interactions (Supplementary Note 1). In this manner, we identified 442 interaction-disrupting SNVs, including 298 disruptive ExAC variants, comprising a network of 4797 SNV-interaction pairs (Fig. 1c). We further validated the quality of our SNV-interaction network by performing Protein Complementation Assay (PCA)[33] in human 293T cells to retest a representative subset of ~400 disrupted and non-disrupted SNV-interactions pairs from our ExAC subset. SNV-disrupted interactions retested at a rate approximate to a negative reference set comprising randomly selected ORF pairs whereas non-disrupted interactions retested at a rate statistically indistinguishable from a positive reference set of literature-established protein interactions[34,35] (Fig. 2a, Supplementary Fig. 1a). Our result remained unchanged when we removed interactions corresponding to highly disruptive SNVs (Supplementary Fig. 1b). Taken together, our PCA retest demonstrated the reproducibility and validated the quality of our Y2H-generated SNV-interaction network.

To examine the influence of allele frequency on disruptive variants, we partitioned our tested ExAC variants across four allele frequency bins, ranging from very rare (MAF ≤ 0.1%) to very common (MAF > 10%) alleles and then calculated the fraction of variants that disrupted one or more protein interactions per MAF bin. We found that the fraction of disruptive variants decreased inversely with increasing allele frequency ($P = 0.0054$ by chi-square test, Fig. 2b), which agrees with expectations[1,4,7]; however, we note that 9.6% of very common variants (MAF > 10%) were still disruptive. Considering that the majority of SNVs found in an individual genome are common[36], this elevated proportion may indicate that disruptive coding variation is markedly widespread across populations. To investigate this more closely, we weighted these MAF-stratified disruption rates by their expected proportions within a typical human genome using the site frequency spectrum for missense variants in ExAC (Supplementary Note 2; see the "Methods" section). In this manner, we determined that given an average of 13,595 missense variants per genome, 1434 (10.5% ± 1.8%; SEM) are expected to disrupt protein interactions, a figure notably higher than indicated by previous estimates (Fig. 2c, Supplementary Tables 1–3). We note, however, that the extent to which interaction disruptions result in cellular phenotypes, particularly for common variants, remains undetermined. Regardless, our results demonstrate that many variants show some degree of functionality, at least within the context of our interaction assays; as such, the genetic background in each individual genome might be far more complex than expected.

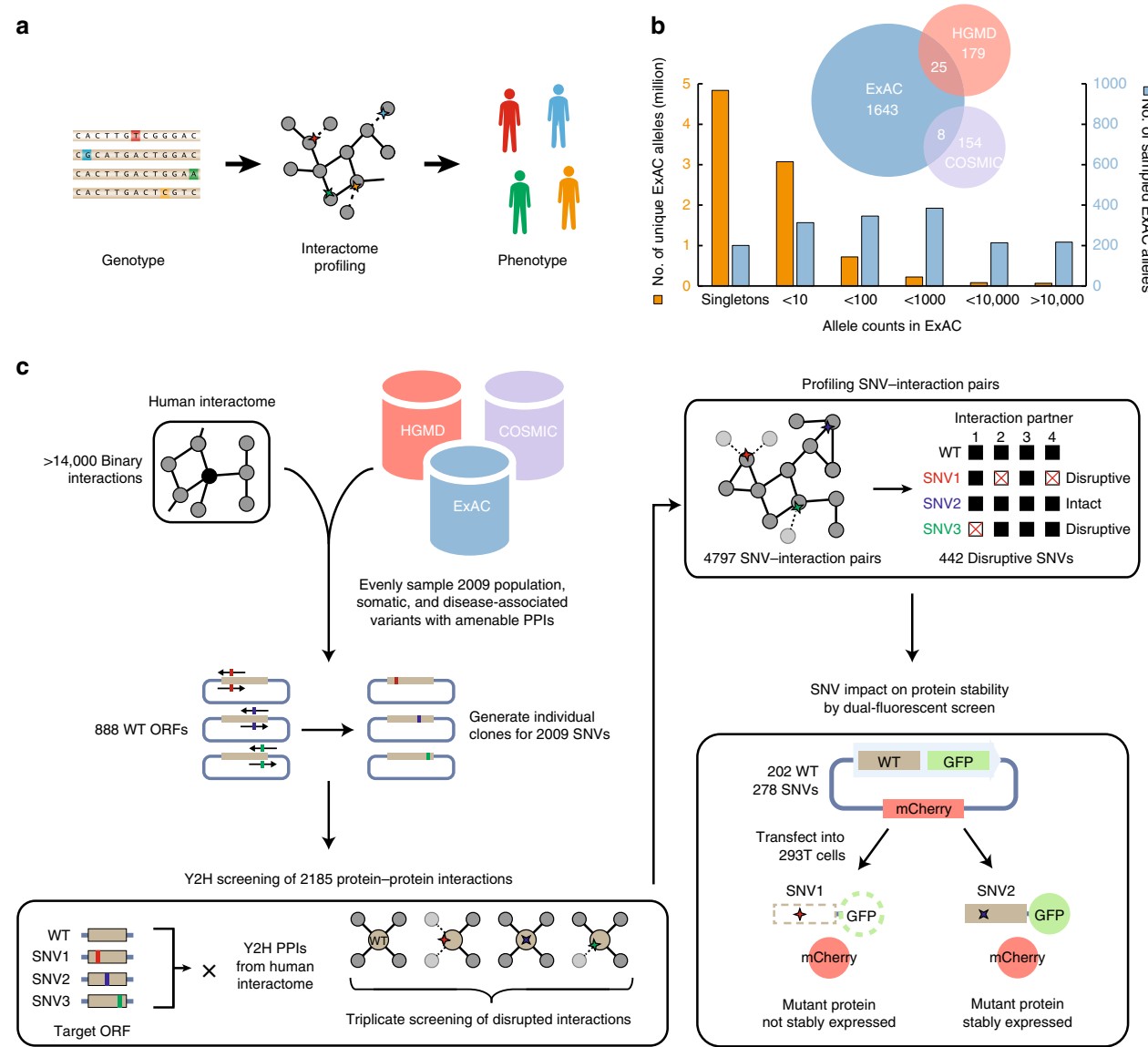

**Fig. 1** A pipeline for surveying the impact of 2009 SNVs on protein–protein interactions. **a** Phenotypic consequences of coding variants in human genotypes can be interpreted as products of protein–protein interaction perturbations in the interactome. **b** Over half of all unique missense variants in ExAC are singletons. To avoid oversampling very rare variants from ExAC, 1676 ExAC variants were selected across a wide range of allele frequencies. 204 disease-associated mutations listed in HGMD and 162 cancer somatic mutations from COSMIC were also examined. **c** Pipeline for testing the functional impact of 2009 SNVs on protein interactions and stability impact of 278 population variants by dual-fluorescence screen

To add further context to our disruption rate analysis, we also determined the fraction of cancer-associated somatic mutations that disrupt interactions and found that 34.8% of somatic mutations located in genes with established roles in cancer progression were disruptive (Fig. 2b; see the "Methods" section). Notably, this fraction decreased significantly to 22.1% for somatic mutations located in all other genes ($P = 0.036$ by one-tailed $Z$-test), a figure comparable to the 20.0% disruption rate observed for very rare (MAF $\leq$ 0.1%) ExAC alleles. In contrast, 52.9% of tested HGMD disease-associated mutations were disruptive (Fig. 2b). Collectively, these trends in disruption rate suggest that driver mutations in oncogenesis may often function by perturbing interactions, as is the case for disease-associated mutations. Therefore, prioritizing disruptive somatic mutations through our interaction perturbation approach may be an effective means to identify potential driver genes and mutations.

The extent to which a mutation is disruptive can also be categorized by measuring the fraction of corresponding protein

interactions disrupted by a particular variant. Accordingly, we first grouped each of our 298 disruptive variants by the number of interactions they perturb (Supplementary Fig. 1c). We observed that 205 of our tested SNVs disrupted only a single interaction (68.8%), while a small fraction of variants (6.7%) disrupted five or more interactions, suggesting that disruptive mutations tend to perturb specific subsets of protein function as opposed to perturbing protein function as a whole. Examining the distribution of disruptive variants across the number of interactions perturbed revealed a similar trend (Supplementary Fig. 1d).

Next, for proteins tested against multiple interaction partners, ExAC variants that leave all interactions intact were categorized as non-disruptive, variants that disrupt a subset of interaction partners were categorized as partially disruptive, and variants that disrupt all tested protein interactions were categorized as null-like (Fig. 2d). Across these three categories, the median allele frequency for tested variants in ExAC decreased significantly from 0.21% for non-disruptive variants to 0.085% for partially

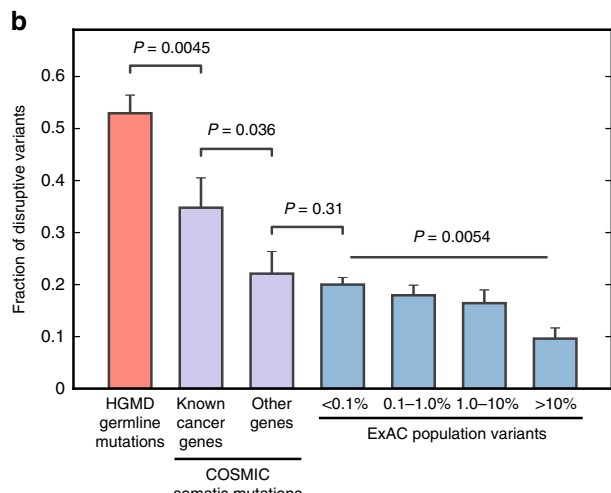

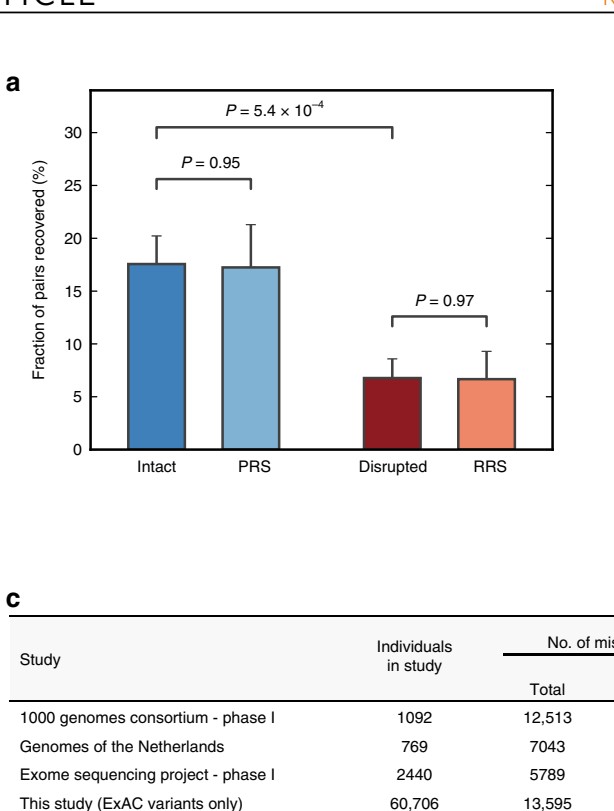

**c**

| Study | Individuals in study | No. of missense variants per individual | | | Method | Citation |
|---|---|---|---|---|---|---|
| | | Total | Functional | % of total | | |
| 1000 genomes consortium - phase I | 1092 | 12,513 | 222 | 1.7 | SFS-deleterious | [7] |
| Genomes of the Netherlands | 769 | 7043 | 484 | 6.9 | PPH2 and GERP>2.0 | [10] |
| Exome sequencing project - phase I | 2440 | 5789 | 318 | 5.5 | SFS-deleterious | [4] |
| This study (ExAC variants only) | 60,706 | 13,595 | 1434 | 10.5 | Interaction perturbation | |

disruptive variants ($P = 0.0067$ by one-tailed $U$-test) then nominally to 0.034% for null-like variants (Fig. 2e), suggesting that partially disruptive and null-like variants are potentially deleterious. Furthermore, we also find that Grantham scores, a biochemical measure quantifying the dissimilarity between amino acid residues[37], for partially disruptive and null-like variants are significantly higher in comparison to non-disruptive variants

(Fig. 2f). Moreover, conservation-based functional prediction algorithms, including PolyPhen-2[27] and MutPred2[38], show significant increases in the likelihood that a variant is deleterious across non-disruptive, partially disruptive, and null-like disruption categories (Fig. 2g, Supplementary Fig. 1e). Taken together, these results show that disruptive variants follow expected patterns of selective constraint and conservation that are

**Fig. 2** Assessing the impact of disruptive alleles on protein function. **a** Fraction of protein pairs recovered by PCA for disrupted and intact interactions in comparison to positive and random reference sets (PRS and RRS). *P* values by one-tailed *Z*-test between disrupted and intact interactions. *P* values by two-tailed *Z*-test for all other comparisons. **b** Fraction of disruptive variants in ExAC (blue) across four allele frequency ranges (i) <0.1%, (ii) 0.1 – 1.0%, (iii) 1.0 – 10%, and (iv) >100%. *P* value by chi-square test. Fraction of disruptive somatic mutations in COSMIC (purple) in known cancer-affiliated genes or other genes and fraction of disruptive germline disease-associated genes from HGMD (red) are also shown. *P* values by one-tailed *Z*-test. **c** Reported number of functional missense variants per individual genome varies extensively across different studies. **d** ExAC variants tested against ≥ 2 interactions further partitioned into three disruption categories. Distribution of **e** allele frequency, **f** Grantham scores, and **g** PolyPhen-2 scores across three disruption categories. Error bars in **a** and **b** indicate + SE of proportion. Thick black bars in **g** are the interquartile range, white dots display the median, and extended thin black lines represent 95% confidence intervals. *P* values in **e**, and **g** by one-tailed *U*-test. *P* values in **f** by two-tailed *U*-test. See also Supplementary Tables 1–3, Supplementary Data 2–4, and Supplementary Fig. 1

characteristic of damaging mutations and imply that these disruptive variants may be functionally relevant in cells.

**Missense variants seldom result in unstable protein expression**. Mutations can disrupt interactions through local perturbations to specific interaction interfaces or by destabilizing protein folding as a whole[22]. To distinguish between these two distinct mechanisms, we developed a dual fluorescence screening assay to survey the impact of interaction-disruptive variants on protein folding. To set up our dual fluorescent screen, we cloned a subset of wild-type (WT) ORFs that are stably expressed when tagged with GFP, as well as their corresponding ExAC variant clones, into a custom GFP-tagged expression vector that co-expresses an untagged mCherry control (see the "Methods" section). We then transfected WT and mutant ORFs into 293T cells to test for mutation-induced changes to protein expression in 96-well plate formats (Fig. 1c). GFP expression levels for transfected WT and mutant samples, normalized with respect to mCherry expression levels, were then used to calculate stability scores for all WT and mutant proteins (Supplementary Note 3). We then grouped variants across stable, moderately stable, and unstable protein expression categories using these stability scores (see the "Methods" section), noting that our three demarcations corresponded well with western blot intensity (Fig. 3a). In this manner, we systematically determined the impact of 278 ExAC variants on protein folding (Fig. 3b).

Mutations that destabilize protein folding should abolish the function of the harboring protein and may likely be depleted within genes that are sensitive to loss-of-function (LoF) mutations as a result. Accordingly, we examined the fraction of variants that occur on genes with pLI ≥ 0.9, a threshold used to define genes that are intolerant to LoF mutations[9], and found that the fraction of variants in LoF-intolerant genes decreased significantly from 27.1% to 13.2% for stable and moderately stable variants, respectively (*P* = 0.018 by one-tailed *U*-test; Fig. 3c). Protein-destabilizing variants also tend to be rare; we observed that median allele frequency decreased from 0.064% for stable protein variants to 0.021% for moderately stable and unstable variants combined (*P* = 0.019 by one-tailed *U*-test; Supplementary Fig. 2a), implying that the destabilized variants uncovered by our protein stability assay are functionally consequential and selectively constrained as a result.

We next investigated the correspondence between protein stability and interaction-disruptive phenotypes by comparing the distribution of stability scores across tested variants from non-disruptive, partially disruptive, and null-like categories. We found that the ratio of mutant-to-WT stability scores is significantly lower for partially disruptive variants than non-disruptive (*P* = 0.0077 by one-tailed *U*-test) and nominally reduced for null-like variants in comparison to partially disruptive variants (Fig. 3d). While destabilized protein expression certainly influences protein interaction perturbations, we found only seven cases (7%) in which an interaction-disruptive variant resulted in unstable

mutant protein expression (Supplementary Fig. 2b). As such, we conclude that most disruptive variants function by inducing local structural perturbations that disrupt specific protein interactions as opposed to destabilizing protein stability as a whole, which agrees with previous studies on disease-associated mutations[19,20]. These results further highlight the importance of dissecting specific interaction disruptions induced by SNVs.

To demonstrate that stably expressed, disruptive variants can be functionally relevant even at common allele frequencies, we characterized a null-like, common variant, A142T (MAF$_{Eur}$ = 9.3%), on the protein AKR7A2 (Fig. 3e). AKR7A2 is an NADPH-dependent aldo-keto reductase that catalyzes the reduction of succinic semialdehyde (SSA) to gamma-hydroxybutyrate (GHB), an important reaction in the degradation pathway for the inhibitory neurotransmitter GABA[39]. Since AKR7A2 is a dimer in solution and A142T disrupts an AKR7A2 interaction with itself, we hypothesized that this mutation might also impact AKR7A2 enzymatic activity. As such, we purified recombinant WT and mutant AKR7A2 protein to test for changes in NADPH-dependent turnover of SSA (see the "Methods" section). Accordingly, we found that $k_{cat}/K_M$ decreased from $1.8 \times 10^7$ min$^{-1}$ M$^{-1}$ for WT protein to $1.0 \times 10^7$ min$^{-1}$ M$^{-1}$ for AKR7A2_A142T (*P* = 0.035 by one-tailed *t*-test, Fig. 3f). In addition to impacting SSA turnover, the A142T mutation is reported to significantly decrease the in vitro metabolism of both doxorubicin and daunorubicin by AKR7A2, which could have important implications in cancer therapy[40]. Moreover, missense mutations that impair ABAT and SSADH activity, enzymes immediately upstream of AKR7A2 (Supplementary Fig. 2c), can result in severe human neurological disorders[41–43]. Hence, we postulate that AKR7A2_A142T may indeed be functionally relevant in genetic backgrounds with lowered ABAT or SSADH activity.

**Disruptive variants are widespread in disease-relevant genes**. We next investigated how disruptive variants are distributed across different gene categories and protein functional sites. We observed comparable enrichment for disruptive variants across disease-associated, cancer-associated, and essential gene sets (Fig. 4a; see the "Methods" section); this enrichment was also comparable to the fraction of disruptive variants found across all genes tested in our SNV-interaction network, excluding highly constrained LoF-intolerant genes (pLI ≥ 0.9), which were significantly depleted for disruptive variants in comparison to other gene sets (Fig. 4a). LoF-intolerant genes correspond well with haploinsufficient genes[9] in which a single mutant copy of a gene is enough to be deleterious. Such genes would be highly sensitive to disruptive variants as a result, potentially explaining the lower fraction of disruptive variants observed in such genes. Notably, duplicate or functionally similar genes can compensate for corresponding proteins impacted by a disruptive mutation. However, we found no enrichment for disruptive variants within a published set of duplicate genes[44] in comparison to non-disruptive

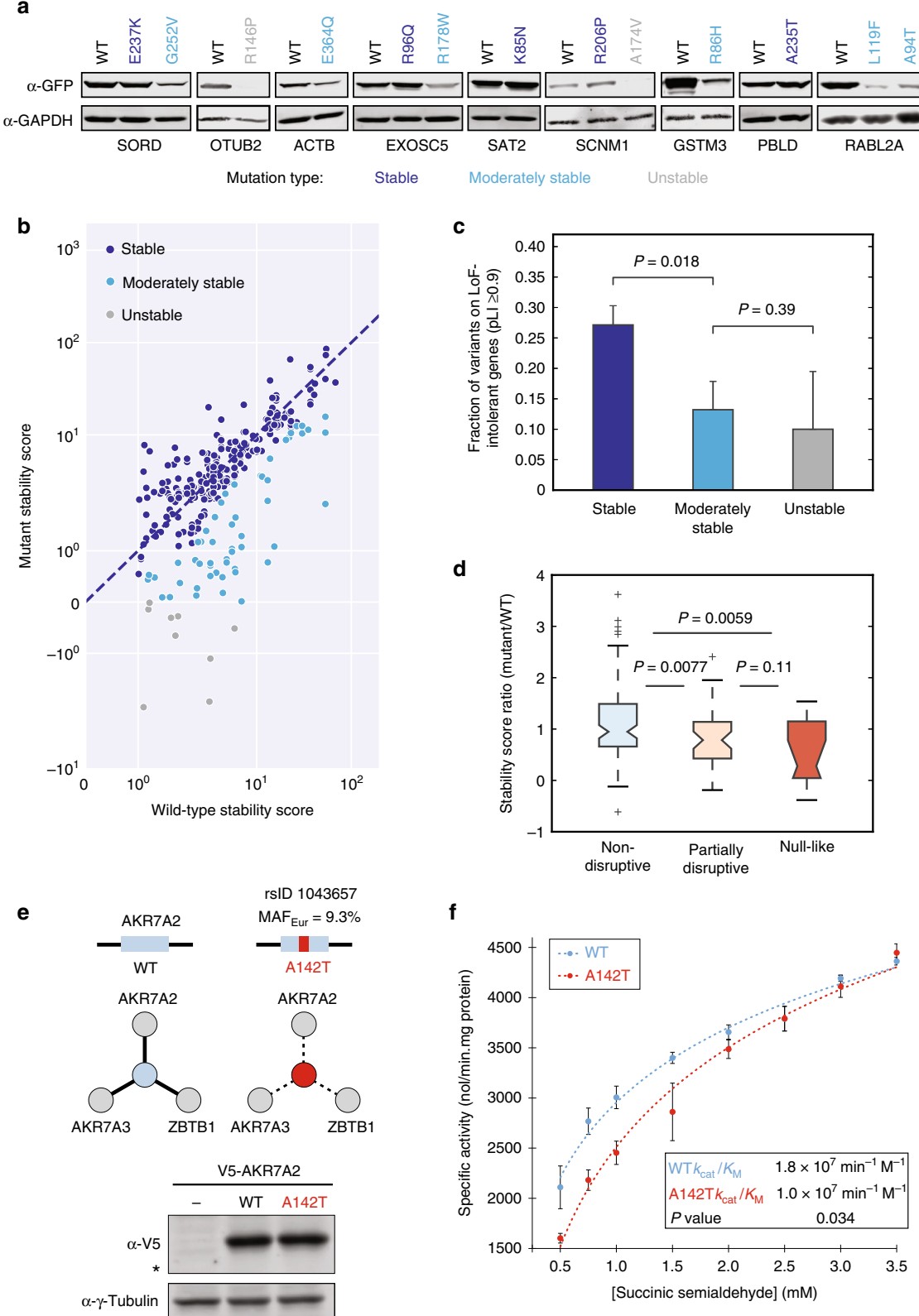

variants (Supplementary Fig. 3a), nor within a custom-generated set of sequence-conserved, functionally similar proteins (Supplementary Fig. 3b; see the "Methods" section). In contrast, a sizable proportion of the disruptive variants in our SNV-interaction network occur in genes relevant to human disease and traits, motivating further exploration into their potential impact.

The structural and genomic loci at which a disruptive variant occurs is strongly indicative of the functional relevance of the mutation. Similar to disease-associated mutations[20], we found that variants located at the interaction interface disrupted interactions significantly more often than variants located away from the interface (19.2% and 5.0%, respectively; $P = 3.9 \times 10^{-5}$ by one-tailed $Z$-test, Fig. 4b; see the "Methods" section). Protein

**Fig. 3** Disruptive population variants seldom result in unstable protein expression. **a** Western blots for representative wild-type:variant pairs across three stability categories detected using α-GFP. α-GAPDH was used as a loading control. **b** DUAL-FLOU protein stability scores for 278 wild-type:variant pairs. Dashed blue line represents 1:1 ratio between stability scores for mutant and wild-type. **c** Fraction of variants residing in LoF-intolerant genes (pLI ≥ 0.9) for stable ($n = 199$), moderately stable ($n = 53$), and unstable ($n = 10$) protein stability categories. **d** Ratio of mutant-to-wild-type stability scores corresponding to non-disruptive ($n = 103$), partially disruptive ($n = 45$), and null-like variants ($n = 12$). **e** Diagram of interactions disrupted by null-like AKR7A2_A142T variant. Cellular expression levels of V5-tagged AKR7A2 was measured by Western blot using α-V5. α-γ-Tubulin was used as a loading control. * indicates 37 kDa marker. **f** In vitro specific activities of purified recombinant AKR7A2 wild-type and A142T using succinic semialdehyde substrate. Fitted curves (dashed lines) are shown for wild-type and A142T. *P* value by one-tailed *t*-test. Error bars indicate ± SE of mean at eight different substrate concentrations. Error bars in **c** and **d** indicate +SE of proportion. *P* values in **c** and **d** by one-tailed *U*-test. See also Supplementary Figs. 2, 6, and 7a

sites corresponding to disruptive variants were also found to be substantially more conserved than those for non-disruptive variants ($P = 1.2 \times 10^{-10}$ by one-tailed *U*-test, Fig. 4c). Purifying selection may also be more specific to disruptive variants at conserved protein sites than non-disruptive variants at equally conserved sites. To demonstrate this, we binned disruptive and non-disruptive variants by their corresponding Jensen–Shannon Divergence (JSD) scores, an amino acid-based metric for conservation, and then compared their mean allele frequency per JSD scoring bin (see the "Methods" section). We found that while allele frequencies for both disruptive and non-disruptive variants were somewhat comparable at low JSD conservation scores, allele frequency for disruptive variants strongly decreased across increasingly stringent JSD cutoffs in comparison to non-disruptive variants (Supplementary Fig. 4a). A similar pattern was also observed using a genomic—as opposed to an amino acid-based—measure for conservation, phyloP[45] (Supplementary Fig. 4b). Therefore, in addition to frequently occurring in disease-relevant genes, disruptive variants are also more likely to occur at functionally important sites in these genes, further implying that a significant fraction of these disruptive variants may be phenotypically relevant.

To complement our exploration of the relationship between conservation and disruptive variation, we also investigated whether disruptive variants tend to occur at genomic regions under positive selection. We applied a test for positive selection based on the distribution of allele frequency around a variant using whole-genome sequencing data from Phase 3 of the 1000 Genomes Project[46] (see the "Methods" section). We observed that genomic regions with disruptive variants exhibit a significant signature of positive selection more often than those with non-disruptive variants. This is the case both within 1000 Genomes continental population groups and globally (Fig. 4d). This result may point to the functional importance of some of the disruptive variants identified here. Therefore, this result also facilitates molecular interpretation of positive selection signals, both in terms of interaction perturbations and by investigating the functions of interacting proteins lost and gained in the presence of these disruptive variants.

**Identifying phenotypic SNVs via matching disruption profiles.** Previous studies have shown that disease-associated mutations often function by perturbing specific protein–protein interactions[19,20,22]. We therefore investigated whether a disruptive population variant with the same interaction impact as a known disease-associated mutation could also result in disease. To do this, we first examined whether pairs of disease-associated mutations that occur on the same gene tend to result in the same interaction perturbations (see the "Methods" section). We found that pairs of disease-associated mutations that share at least one or more disrupted interactions resulted in the same disease significantly more often than mutations that did not share any disrupted interactions (0.738 to 0.630, respectively; $P = 8.5 \times 10^{-4}$ by one-tailed *Z*-test, Fig. 4e). This trend persisted when mutation

pairs shared two or more disrupted interactions in comparison to no shared disrupted interactions (0.760 to 0.630, respectively; $P = 0.018$ by one-tailed *Z*-test, Fig. 4e). This result therefore suggests that shared interaction disruption profiles may be an informative approach to prioritizing candidate disease-associated mutations.

To demonstrate how pairs of disease-associated mutations on the same gene with matching disruption profiles can result in the same disease, we highlight three disease-associated mutations on SMAD4 (Fig. 4f), a crucial protein in the TGFβ/SMAD signaling pathway. Two mutations on SMAD4, E330K, and G352R, are associated with juvenile polyposis[47,48], while a third mutation, N13S, results in a clinically distinct disease, pulmonary arterial hypertension[49]. We observed that E330K and G352R cluster together in three-dimensional space near the SMAD4–SMAD3 interaction interface (Fig. 4g). N13S, in contrast, appears positioned away from E330K and G352R near the N-terminus of SMAD4. In agreement with the proximal clustering of E330K and G352R near the SMAD4–SMAD3 interaction interface, both mutations disrupted the SMAD4 interaction with SMAD3 in addition to disrupting the SMAD4–SMAD9 interaction (Fig. 4f). These SMAD protein disruption results agree with previous evidence implicating the TGFβ/SMAD signaling pathway in the formation of juvenile polyposis[50,51]. In contrast, the N13S mutation left SMAD4 interactions with SMAD3 and SMAD9 intact, which agrees with a previous study that found no evidence that N13S alters SMAD-mediated signaling[49].

With this example as a template, we then explored cases in which both an ExAC variant and a known disease-associated mutation shared the same disruption profile with the goal of determining whether the population variant can result in the same disease phenotype. To do this, we tested two mutations with matching disruption profiles on the protein PSPH (Fig. 5a): (i) T152I, a rare variant (MAF = 0.10%) in ExAC that disrupts an interaction with itself and (ii) D32N, a disease-associated mutation that also disrupts an interaction with itself and causes phosphoserine phosphatase deficiency in a compound heterozygous individual with two deleterious PSPH mutations[52]. An additional PSPH non-disruptive rare variant, T149M, was included as a control. Since PSPH exists as a dimer in solution and can aggregate when mutations that interfere with dimerization are introduced[53], we reasoned that mutations that disrupt this dimerization may also reduce PSPH enzymatic activity. We therefore purified recombinant WT, D32N, T152I, and T149M PSPH proteins and measured for changes in phosphatase activity for PSPH mutants relative to WT using a malachite green assay. Our in vitro assays revealed that T152I significantly reduced PSPH phosphatase activity to 59.2% ± 4.3% relative to WT (SEM; $P = 0.0010$ by one-tailed *t*-test; $n = 3$), which nearly matched the D32N reduction in activity (60.0% ± 3.3%, $P = 6.6 \times 10^{-4}$ by one-tailed *t*-test compared to WT; $n = 3$). In contrast, T149M showed no significant change in enzymatic activity relative to WT ($P = 0.19$ by one-tailed *t*-test, Fig. 5b). Because phosphoserine phosphatase deficiency is a recessively inherited

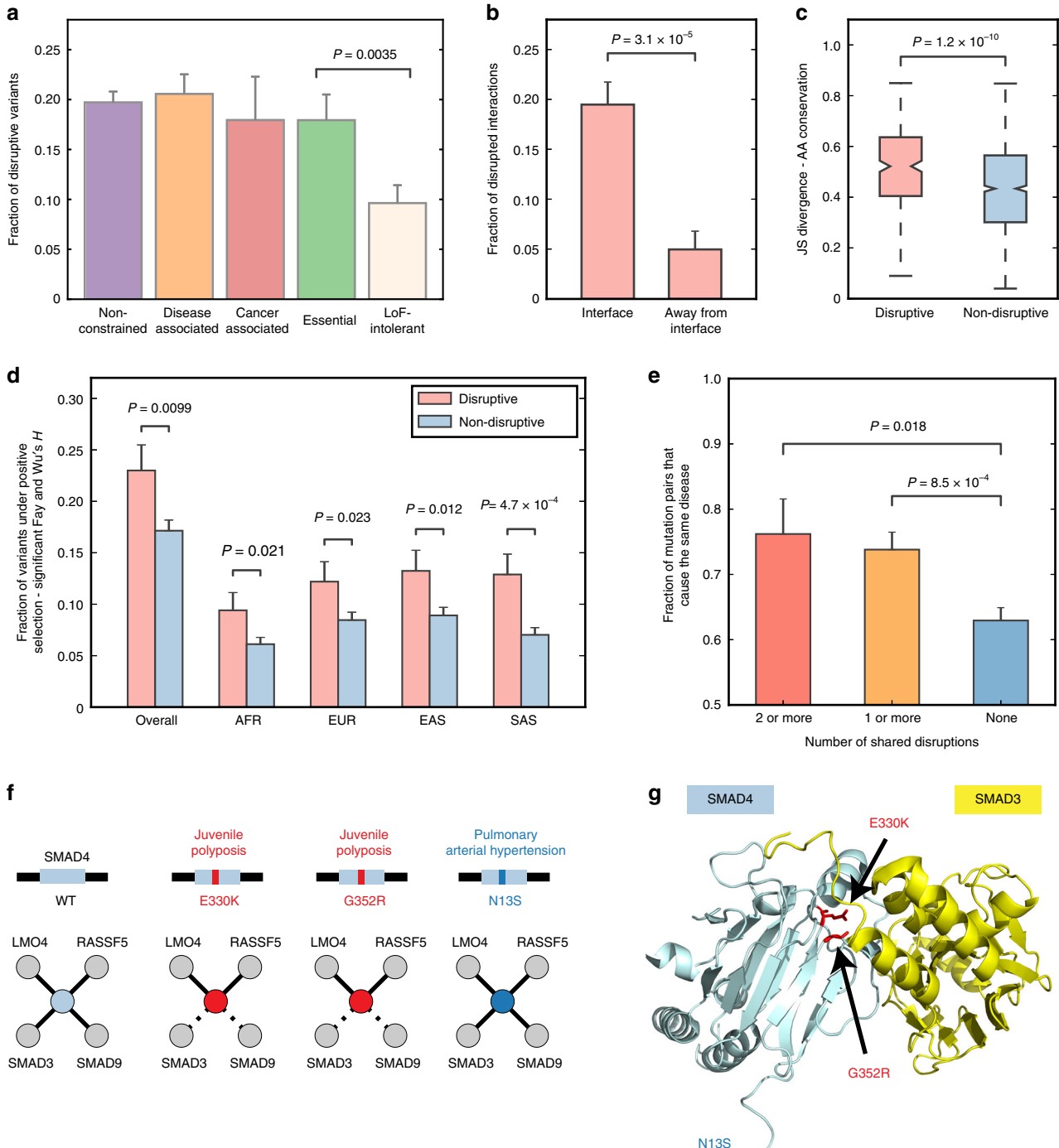

**Fig. 4** Disruptive variants occur in important gene groups and at conserved genomic sites. **a** Fraction of disruptive variants that occur in non-constrained ($n = 1349$), disease-associated ($n = 423$), cancer-associated ($n = 78$), essential ($n = 223$), or LoF-intolerant genes ($n = 270$). **b** Fraction of interactions disrupted by variants that occur on interface residues or interface domains ($n = 307$) in comparison to interactions disrupted by variants that occur away from interaction interfaces ($n = 41$). **c** Distribution of Jensen–Shannon Divergence scores for amino acid residues at sites corresponding to disruptive and non-disruptive variants. Larger scores indicate more conserved sites. **d** Fraction of disruptive variants found in genomic regions where Fay and Wu's $H$ is significant measured across four different population groups and across overall population. **e** Fraction of mutations pairs that lead to the same disease for germline mutations that share two or more disrupted interactions ($n = 42$), share one or more disrupted interactions ($n = 271$), or do not share disrupted interactions ($n = 599$). **f** Schematic of interaction disruption profiles for SMAD4 disease-associated mutations E330K, G352R, and N13S. Corresponding disease names are labeled. **g** Co-crystal structure of SMAD4–SMAD3 interacting proteins (PDB ID: 1U7F). Disease-associated mutations are labeled. Structure covers SMAD4 residues 315-546 and therefore N13S mutation is not represented on this structure. Error bars in **a**, **b**, **d**, and **e** indicate +SE of proportion. $P$ values in **a**, **b**, **d**, and **e** by one-tailed $Z$-test. $P$ value in **c** by one-tailed $U$-test. See also Supplementary Fig. 4

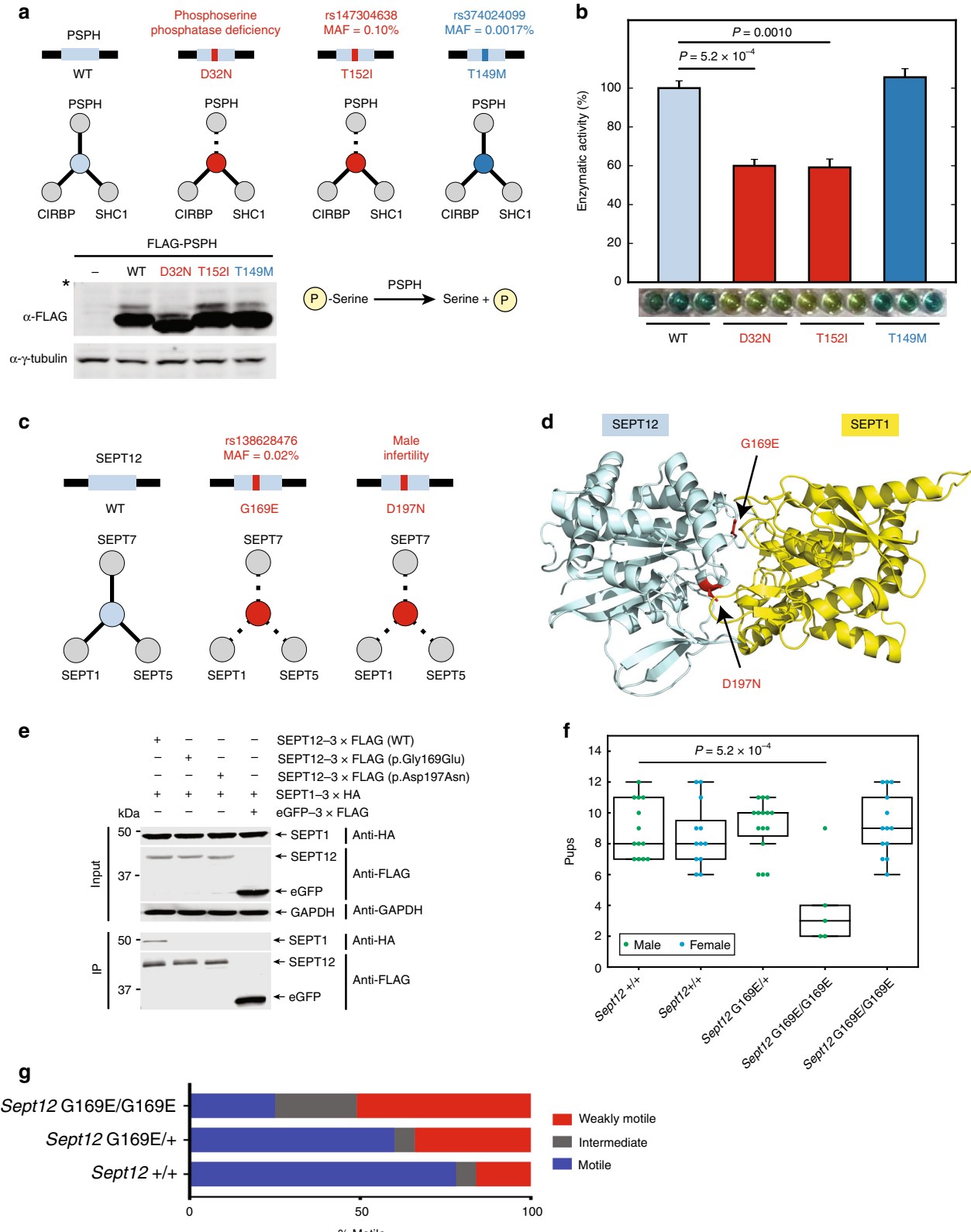

condition[52], our findings suggest that T152I may lead to the same disease phenotype in homozygous or compound heterozygous individuals.

To further demonstrate how potential physiologically relevant mutations can be identified using shared disruption profiles, we also characterized a pair of disruptive mutations on the GTPase, SEPT12: G169E (MAF = 0.02%), a rare variant not known to

associate with any disease phenotypes, and D197N, an infertility-causing mutation in men[54]. Both mutations perturbed interactions with SEPT7 and SEPT2 subgroup proteins, SEPT1 and SEPT5 (Fig. 5c). These perturbations are particularly relevant because SEPT12 is known to interact with other septin proteins found in the SEPT2, SEPT6, and SEPT7 protein subgroups to form a filamentous structure at the sperm annulus[55–57].

**Fig. 5** Prioritizing candidate disease-associated mutations through shared disruption profiles. **a** Schematic of interaction disruption profiles for disease-associated mutation D32N and rare variants T152I and T149M. Stable expression of FLAG-tagged wild-type and mutant PSPH proteins was validated by Western blot using α-FLAG. α-γ-Tubulin was used as a loading control. A brief diagram of PSPH phosphatase activity is shown. * indicates 37 kDa marker **b** Enzymatic activity of purified recombinant wild-type and mutant PSPH proteins using phosphoserine substrate was measured in vitro using a malachite green assay performed in triplicate. Enzymatic activities for PSPH mutants are shown in proportion to wild-type activity. Error bars indicate +SE of mean. *P* value by one-tailed *t*-test. **c** Schematic of interaction disruption profiles for SEPT12 rare variant G169E and disease-associated mutation D197N. **d** Homology model of SEPT12–SEPT1 interaction. PDB ID 5CYO chains A and B used as template. Disruptive mutations on interaction interface are labeled. **e** Disruption of SEPT12 interaction with SEPT1 by G169E and D197N was validated by co-IP. SEPT12 bait proteins were detected using α-FLAG. SEPT1 prey was detected using α-HA. α-GAPDH was used as a loading control. **f** Fertility tests of 2–6-month-old WT ($n = 2$ males, avg $= 8.9 \pm 0.51$; $n = 2$ females, avg $= 8.6 \pm 0.61$) and Sept12$^{G169E/G169E}$ ($n = 3$ males, avg $= 4.0 \pm 1.3$; $n = 2$ females, avg $= 9.2 \pm 0.57$) mice bred to age-matched controls. Litter sizes were recorded. Green = males. Blue = females. All comparisons are not significant except for male WT vs. male Sept12$^{G169E/G169E}$ ($P = 0.00052$; by two-tailed *t*-test). **g** Assessment of sperm motility of WT ($n = 2$, sperm $= 166$), Sept12$^{G169E/+}$ ($n = 4$, sperm $= 484$), and Sept12$^{G169E/G169E}$ ($n = 3$, sperm $= 416$) mice. See also Supplementary Figs. 7b, 8, and 9

Moreover, the infertility-causing mutation SEPT12_D197N has been previously shown to perturb interactions with these same septin subgroup proteins, resulting in a disorganized sperm annulus and poor sperm motility in a mouse model for D197N[57]. Lastly, using homology modeling, we observed that both G169E and D197N mutations occur at SEPT12 interaction interface residues with SEPT1 (Fig. 5d) and confirmed that both mutations disrupt the SEPT12–SEPT1 interaction without reducing protein stability in 293T cells (Fig. 5e). These results demonstrate that these mutations function by specifically perturbing SEPT12 protein–protein interactions, as opposed to disrupting SEPT12 stability as a whole.

We then investigated whether these matching SEPT12 molecular phenotypes result in corresponding organismal phenotypes by generating Sept12$^{G169E}$ mice using a CRISPR-editing approach[58]. We found that homozygous Sept12$^{G169E}$ males were subfertile in comparison to WT males (Fig. 5f). Notably, sperm from homozygous Sept12$^{G169E}$ males exhibited poor motility (Fig. 5g), a phenotype also reported for Sept12$^{D197N}$ male mice[57]. Our observations of poor sperm motility and subfertility in homozygous Sept12$^{G169E}$ male mice suggest that SEPT12_G169E may deleteriously impact fertility in men homozygous for this mutation, although we note that no individuals homozygous for SEPT12_G169E have been reported in ExAC. Taken together with our in vitro data, these results also demonstrate how shared disruption profiles can be used to prioritize candidate disease-associated mutations.

## Discussion

Disentangling the phenotypic impact of functional missense mutations from benign mutations has proven to be uniquely challenging[59,60]. Conventions for determining which missense mutations are functional vary widely[4,7,10], as do their genome-wide estimates for the number of functional coding mutations per individual (Fig. 2c). These inconsistencies are problematic, since accurate measurements of the impact of SNVs on protein functions are essential for generating concrete hypotheses about disease etiology based on molecular mechanisms[21]. Therefore, in the absence of a consensus metric for assessing the functional impact of missense mutations across a large set of proteins, we directly measured the impact of 1676 missense ExAC-listed population variants (811 with MAF > 0.1%) across 4109 protein-variant interaction pairs and identified 298 disruptive variants affecting 669 human protein interactions. In this manner, we have constructed an unbiased resource to examine the relationships between the population, genetic, and evolutionary characteristics of SNVs and their functional impacts genome-wide.

By weighting our measured disruption rates against their expected proportions per individual genome, we further determined that 10.5% of missense variants per individual are expected

to be disruptive. It should be noted that, like any high-throughput assay, Y2H cannot detect all interactions for a given protein. If we were able to detect more interactions, we would likely discover more interaction disruptions. Therefore, this 10.5% figure represents only a lower-bound estimate for the number of disruptive missense variants per individual. Furthermore, considering that interaction perturbations are just one way in which mutations can perturb protein function, genome-wide surveys for other types of activities (e.g., enzymatic activities, transcription factors binding to DNA, etc.) may reveal that functional variants, at least at the molecular-level, are even more widespread than suggested here. Finally, we note that literature-curated sources are not appropriate for reproducing the analyses presented here because of their strong biases to synthetic and very rare mutations. Even literature-curated mutations listed at appreciable allele frequencies may be inappropriate, since such mutations are often selected because of their known disease associations. For example, a recent study comprehensively collected the impact of 7955 mutations on human protein interactions published in the literature[61]; however, only 161 of these mutations were without disease annotations and listed in ExAC, of which 49 occurred at appreciable frequencies (MAF > 0.1%).

The genetic and genomic context in which a variant occurs is crucial for properly interpreting the functional impact that a disruptive mutation may have. While haplosufficiency likely mitigates the impact of numerous disruptive variants, an individual already harboring one disruptive variant can become sensitized to the consequences of subsequent mutations in the same gene or pathway. For example, we identified a null-like, common variant, A142T, on the protein AKR7A2 that significantly reduces its enzymatic activity relative to WT (Fig. 3g). This mutation alone likely has a minimal impact on fitness; however, mutations to enzymes immediately upstream to AKR7A2, particularly ABAT and SSADH (Supplementary Fig. 2b), can result in severe neurological disorders[41–43]. Co-occurrence of AKR7A2_A142T with similarly disruptive mutations in ABAT or SSADH could therefore result in a neurological disorder that would not otherwise occur in an individual harboring only a single disruptive mutation. Such relationships are frequent in complex disease, including cancer and heart disease, which unlike Mendelian mutations, require multiple mutations on more than one gene to cause a disorder. Each disease-associated mutation in complex disease therefore contributes a certain measure of disease risk that can be quantified by a GWAS, and some authors consider these effects to be approximately additive across loci[62]. Measuring how one mutation modulates the impact of another is challenging; however, measuring which mutations are individually functional is a crucial first step. Hence, we anticipate that our SNV-interaction network will serve as a pivotal framework for defining the epistatic relationships that modulate the impact of disruptive

variants, particularly for partially penetrant variants that only result in disease in certain genetic backgrounds (Supplementary Note 4).

Several methods to experimentally measure the impact of coding mutations at large scales have been recently reported[20,63–65]. The depth of proteins, variants, and interactions presented here complements these previous methods well. For example, Fields and Shendure developed a massively parallel single-amino-acid mutagenesis pipeline, named PALS, that can generate nearly all potential singleton mutations possible for a particular gene of interest[63]. This impressive depth makes PALS an excellent method for studying extensive variation in a single protein, most notably TP53[63] and BRCA1[66], but remains to be optimized for studying variation across a large set of unique genes. In contrast, our mutagenesis approach allowed us to survey > 2000 mutations across 847 unique genes. Similarly, while Y2H is widely used for characterizing the impact of mutations on protein–protein interactions[19,20], several derivatives for detecting perturbations by Y2H exist. For instance, Stelzl and colleagues developed the Int-Seq platform for probing protein–protein interaction disruptions using a Reverse Two-Hybrid (R2H) approach[65]. While this R2H approach increases assay sensitivity, a R2H reference interactome is not yet available, limiting the coverage of this approach to a handful of interactions.

Interaction perturbations constitute only a particular subset of the variety of ways in which mutations can impair protein function. Continued efforts to survey all potential manners in which molecular-level perturbations can alter cellular and organismal phenotypes are needed to properly understand the impact of mutations on human health. Although our experimental framework was not designed to find potentially causal variants driving GWAS phenotypes (Supplementary Fig. 5; see the "Methods" section), experimental frameworks that can differentiate functional variants from those that are non-functional will be key to identifying causal variants in common disease as well as for characterizing SNVs that alter drug–protein interactions (Supplementary Note 5; see the "Methods" section). Towards this goal, the genetic, protein interaction, and population-level insights presented here may represent a pivotal step forward to an improved understanding of the evolutionary forces that shape the human genome and protein function.

## Methods

**Selecting SNVs from ExAC, HGMD, and COSMIC databases.** Population variants encoding for missense mutations were selected from ExAC release 0.3.1[9]. Unless a specific subpopulation is listed, all reported allele frequencies and allele frequency-derived calculations refer to allele frequency across all ExAC populations. Disease-associated missense mutations were obtained from HGMD (Public release version, 2014). Cancer-associated somatic missense mutations were selected from COSMIC version 84. For all three datasets, we required that (i) mutations reside on genes in either hORFeome v8.1[67] or v5.1[68], (ii) corresponded with one or more high-throughput Y2H-testable protein–protein interactions (Supplementary Note 1), and (iii) for ExAC variants, achieved a PASS filter status. We mapped each RefSeq transcript from ExAC to an appropriate ORF in our library by looking at the top BLAXTX candidate with an E-value ≤ 0.001. We verified that this was a representative ORF for our mutation by performing EDNAFULL matrix pairwise alignment using EMBOSS Stretcher. Valid representative ORFs had to be identical within a 31 amino acid window centered on the position of interest for mutagenesis. Beyond local identity, ORFs were required to have more than 95% global identity, or be an exact subset of the transcript, spanning at least a third of the query transcript.

Since over half of all variants in ExAC are singletons, to avoid oversampling rare alleles, we selected between 200 and 400 variants across six mutually exclusive allele count bins of 1, <10, <100, <1000, <10,000, and >10,000 for a total of 1676 ExAC alleles (Fig. 1b). In each bin, we randomly selected variants on genes with Y2H-testable interactions. To minimize gene bias, we selected an average of two variants per gene. 204 HGMD mutations listed as DM (disease-causing mutations) were selected in accordance to criteria detailed in ref. [20] but expanded to test across all amenable Y2H protein–protein interactions. 162 COSMIC mutations among 110 different genes with available hORFeome clones were also tested across all amenable Y2H protein–protein interactions. Genes listed in the Cancer Gene

Census (v84) and listed as a Tier 1 known drivers in IntOGen (2016.5) were designated as Known cancer genes. Genes not listed in the Cancer Gene Census and not listed as a driver in IntOGen were designated as Other genes (Fig. 2b).

**Large-scale cloning of SNVs through Clone-seq pipeline.** Single colony-derived mutant clones were constructed using Clone-seq[20], a high-throughput mutagenesis and next-generation sequencing platform. In brief, WT clones were picked from hORFeome clones and served as templates for site-directed mutagenesis performed in 96-well plates using site-specific mutagenesis primers (Eurofins). Primers for mutagenesis were designed using the webtool primer.yulab.org, and a list of all primers used in this study is provided in Supplementary Data 1. To minimize sequencing artifacts, PCR was limited to 18 cycles using Phusion polymerase (NEB, M0530). PCR products were digested overnight with DpnI (NEB, R0176) then transformed into competent bacteria cells to isolate single colonies. Up to four colonies per individual mutagenesis reaction were then hand-picked and arrayed into 96-well plates and incubated for 21 h at 37 °C under constant vibration. After incubation, glycerol stocks were generated; clones were then pooled into independent bacterial pools. An additional maxiprepped bacterial pool comprising only WT DNA templates corresponding to each mutagenesis PCR reaction was also prepared. Maxiprepped clonal DNA from each bacterial pool was then combined through multiplexing (NEB, E7335) and sequenced in a single 1 × 75 single-end Illumina NextSeq run. Properly mutated clones which differed from their sequenced WT templates only by the desired single base-pair mutation—and nowhere else—were then identified by next-generation sequencing analysis and recovered from their corresponding single colony glycerol stocks.

**Identifying successfully mutated clones.** After de-multiplexing, mapped reads corresponding to the generated pools (wildtype plus up to four mutant pools) were mapped to genes of interest using the BWA mem algorithm (bwa mem -a -t 12 < reference > < reads > ). In order to detect both the desired variant as well as undesired off-target mutations, we first obtained the read counts for each allele (A, T, C, G, insertion, or deletion) for all positions in the clones. Using these read counts we calculated the score for a given position, pos, containing a mutation from the wildtype allele, WT, to a mutant allele, Mut, as follows:

$$\text{Score}(\text{WT}, \text{pos}, \text{Mut}) = \frac{\text{Observed}_{\text{Mut,pos}}}{\text{Expected}_{\text{Mut,pos}}} \qquad (1)$$

where $\text{Observed}_{\text{Mut,pos}}$ is the observed fraction of reads at position pos matching allele Mut and $\text{Expected}_{\text{Mut,pos}}$ is the fraction of reads at position pos matching allele Mut that we would expect to see if it the mutation in question had indeed occurred. We define this fraction as:

$$\text{Expected}_{\text{Mut,pos}} = \frac{1}{\text{TotalMutations}} + (\text{TotalMutations} - 1) \cdot \text{SeqErr(pos)} \\ - (\text{Alleles} - 1) \cdot \text{SeqErr(pos)} \qquad (2)$$

where TotalMutations is the total number of mutants attempted for a particular ORF (i.e. the number of copies of the ORF included in the pool), SeqErr(pos) refers to the inherent sequencing error, and Alleles is the total number of alleles.

To explain further, assuming that all clones for a particular gene contribute a similar number of reads, we expect that if one of the clones for a gene contains a mutation to the Mut allele at position pos, we should see $\frac{1}{\text{TotalMutations}}$ fraction of the reads match the Mut allele. Due to sequencing errors, we expect the true fraction observed to deviate slightly from this base fraction. We first add a term for the fraction of Mut alleles that we expect to see as a result of sequencing errors in the other non-mutant clones for the gene. Second, we subtract a term for sequencing errors in the mutant clone converting the Mut allele to any of the (Alleles-1) other alleles. We define the sequencing error as the average fraction of non-WT bases observed in the 10 closest positions that were not targeted for mutagenesis.

Based on comparisons to Sanger sequencing results, we set a threshold of Score(WT, pos, Mut) ≥ 0.5 to call true mutations. In identifying successful instances of site-directed mutagenesis, we first checked for the presence of the desired mutation using this score threshold. Using the scores for all other positions along the clone, we then screened each successful mutant for the presence of any other unwanted mutations that may have been introduced as PCR artifacts. Any clones containing unwanted mutations were removed, and the remaining clones were sorted using a combination of their desired mutation score, maximum undesired mutation score, sequencing coverage, and sequencing quality.

**Calculating proportion of functional mutations exome-wide.** The total number of missense variants in ExAC release 0.3.1, diagrammed in Fig. 1b, was determined by summing the adjusted allele count found in the ExAC database for all variants annotated as missense_variant in at least one transcript. The number of functional mutations was calculated by multiplying the mean disruption rate per individual (Supplementary Note 2) by the total number of missense variants in ExAC.

The total number of missense variants in the 1000 Genomes Consortium—Phase I, Genomes of the Netherlands, and Exome Sequencing Project—Phase I were obtained from refs. [4,7,10], respectively. Calculations for the number of functional missense mutations from each source are annotated in Supplementary

Tables 1–3. We note that the number of functional mutations by mutation type was not reported for ESP variants in ref. [4]. As such, functional nonsynonymous mutations, including nonsense variants, were instead reported for ESP—Phase I. We expect the proportions of functional missense variants for ESP, 5.5% and 10.0% using conservative and liberal criteria counts listed in ref. [4], respectively, (Supplementary Table 3), to be slight overestimates as a result.

**Profiling disrupted protein interactions through Y2H**. Clone-seq-identified mutant clones were transferred into Y2H vectors pDEST-AD and pDEST-DB by Gateway LR reactions then transformed into *MAT***a** Y8800 and *MAT*α Y8930, respectively. All DB-ORF *MAT*α transformants, including WT ORFs, were then mated against corresponding WT and mutant AD-ORF *MAT***a** transformants in a pairwise orientation using automated 96-well procedures to inoculate AD-ORF and DB-ORF yeast cultures followed by mating on YEPD agar plates. All DB-ORF yeast cultures were also mated against *MAT***a** yeast transformed with empty pDEST-AD vector to screen for autoactivators. After overnight incubation at 30 °C, yeast were replica-plated onto selective Synthetic Complete agar media lacking leucine and tryptophan (SC-Leu-Trp) to select for mated diploid yeast then incubated again overnight at 30 °C. Diploid yeast were then replica-plated onto SC-Leu-Trp agar plates also lacking histidine and supplemented with 1 mM of 3-amino-1,2,4-triazole (SC-Leu-Trp-His+3AT), as well as SC-Leu-Trp agar plates lacking adenine (SC-Leu-Trp-Ade). After overnight incubation at 30 °C, plates were replica-cleaned and incubated again for three days at 30 °C.

Disrupted protein–protein interactions were identified as follows: (1) mutated protein reduces growth by at least 50% relative to WT interaction as benchmarked by twofold serial dilution experiments, (2) neither WT or mutant DB-ORFs are autoactivators and, (3) reduced growth phenotype reproduces across three screens. A mutation was scored as disruptive if one or more corresponding protein-protein interactions were disrupted and was scored as non-disrupted if otherwise. Mutations tested against two or more interactions partners were further categorized as non-disruptive, partially disruptive, and null-like if no tested interactions were perturbed, some tested interactions were perturbed, or all tested interactions were perturbed, respectively. PSPH interactions with CIRBP and SHC1 were detected using PCA. No significant change in PCA signal intensity was detected between any WT or mutant PSPH interactions with CIRBP and SHC1 and therefore all mutant PSPH interactions with CIRBP and SHC1 were scored as non-disrupted. Interaction disruption data for all tested ExAC variants, COSMIC somatic mutations, and HGMD disease-associated mutations can be found in Supplementary Data 2–4, respectively.

**DUAL-FLUO assay to measure SNV impact on protein stability**. In order to screen for variants that destabilize protein expression, we first screened for stably expressed GFP-tagged WT proteins. To do this, we transferred WT ORFs into pDEST-DUAL by Gateway LR reactions. HEK293T cells (ATCC, CRL-3216) were then seeded onto black 96-well flat-bottom dishes (Costar, 3603). HEK293T cells were maintained in complete DMEM medium supplemented with 10% FBS. All cell incubation steps were performed at 37 °C under air with 5% CO2. Cells were grown to 60% confluency then co-transfected using 150 ng sample DNA in pDEST-DUAL and 1.0 μL of 1 mg/mL PEI (Polysciences Inc, 23966) mixed thoroughly with 20 μL OptiMEM (Gibco, 31985-062). Four replicates of empty pDEST-DUAL and four replicates of empty pcDNA-DEST47 were also transfected per 96-well plate as positive controls for mCherry expression and negative controls for GFP expression, respectively. After 72 h incubation, stably expressed WT GFP-tagged proteins were identified using a Tecan M1000 plate reader. Samples that resulted in GFP and mCherry expression significantly above background were confirmed by automated fluorescence microscopy using an ImageXpress system. In this manner, we identified 202 WT genes corresponding to 278 ExAC variants. Single clones for ExAC variants were then transferred into pDEST-DUAL by Gateway LR reactions for further screening.

WT and mutant ORF pairs in pDEST-DUAL were transfected into 293T cells in the same fashion as described for our first WT screen, including eight total pDEST-DUAL and pcDNA-DEST47 controls per plate. Mutant ORFs corresponding to a particular WT ORF were always partitioned onto the same plate. After 72 h incubation, GFP and mCherry fluorescence readings using a Tecan M1000 plate reader were measured for all samples and imaged by automated fluorescence microscopy using an ImageXpress system. Mutant proteins were then processed into stable, moderately stable, and unstable categories of protein expression as follows: if the ratio between mutant and WT stability scores fell below 0.5, indicating that the mutant protein is still expressed but at markedly reduced levels, we categorized the mutant protein as moderately stable. If mutant protein expression dropped below plate reader detection thresholds, as indicated by a mutant stability score < 0, we instead categorized the mutant protein as unstable. Mutant proteins above both thresholds are scored as stable (Supplementary Note 3). Stability data for tested ExAC variants are reported in Supplementary Data 5.

**Retesting disrupted and non-disrupted interactions by PCA**. To confirm that variant-disrupted protein–protein interactions are reproducible across a different assay, we systemically selected a subset of Y2H-tested mutant protein interactions for retesting by PCA. We provide background for applying this method towards retesting interaction perturbations in Supplementary Note 6. Bait ORFs in pDONR223 for disruptive and non-disruptive variants were transferred into F1 Venus fragments while prey ORFs for corresponding interaction partners were transferred into F2 Venus fragments using Gateway LR reactions for a total of 192 Y2H-disrupted mutant interaction pairs and 205 non-disrupted Y2H mutant interaction pairs. Bait and prey ORF pairs from both sets were then randomly scrambled across 87 PRS and 90 RRS ORF pairs previously described in refs. [34,35] to minimize detection bias across different 96-well plates. As a quality control measure, interaction pairs in which either a bait or prey ORF did not amplify by PCR using F1 Venus- or F2 Venus-specific primers, respectively, were removed from PCA analysis.

To perform PCA, HEK293T cells (ATCC, CRL-3216) were seeded onto black 96-well flat-bottom dishes (Costar, 3603). HEK293T cells were maintained in complete DMEM medium supplemented with 10% FBS and incubated at 37 °C under air with 5% CO2. Cells were grown to 60–70% confluency then co-transfected using 100 ng bait vector plus 100 ng prey vector with 1.0 μL of 1 mg/mL PEI (Polysciences Inc., 23966) mixed thoroughly with 20 μL OptiMEM (Gibco, 31985-062) per transfection. After 72 h incubation at 37 °C, a Tecan M1000 plate reader was used to measure PCA fluorescence (excitation = 514 nm; excitation = 527 nm) for all samples. A manually adjusted gain was applied to ensure all measurements were performed within a constant linear range. Detection thresholds were selected such that ORF pairs resulting in a signal greater than the threshold were scored as detected while scores that fell below the threshold were scored as undetected. The fraction of recovered pairs represents the proportion of ORF pairs that scored above a given threshold over the total set of ORF pairs tested per category. Results are reported in Supplementary Data 6.

**Constructing vectors for DUAL-FLUO screen and Western blot**. Gateway LR reactions were used to transfer ORFs into mammalian expression vectors. The pDEST-DUAL vector for our dual-fluorescence screen was constructed by inserting an mCherry cassette independently driven by a minCMV promoter into pcDNA-DEST47 (Invitrogen, 12281-010), which features a C-terminal GFP tag. PSPH WT, D32N, T152I, and T149M were transferred into a pQCXIP (ClonTech, 631516) vector modified to include a Gateway cassette featuring a C-terminal 3 × FLAG tag. SEPT12 WT, G169E, and D197N were transferred also into this same modified pQCXIP 3 × FLAG vector. SEPT1 was transferred into a modified pcDNA3.1 (Invitrogen, V79020) vector featuring a C-terminal 3 × HA tag. AKR7A2 WT and A142T were transferred into pcDNA-DEST40, which includes a V5 tag (Invitrogen, 12274-015).

**Cell culture for Western blotting**. HEK293T cells (ATCC, CRL-3216) were maintained in complete DMEM medium supplemented with 10% FBS and incubated at 37 °C under air with 5% CO2. Cells were grown in 6-well dishes to 70–80% confluency then transfected using 2 μg of vector with 10 μL of 1 mg/mL PEI (Polysciences Inc., 23966) mixed thoroughly with 150 μL OptiMEM (Gibco, 31985-062). After 24 h incubation, cells were gently washed three times in 1× PBS and then resuspended in 200 μL cell lysis buffer [10 mM Tris–Cl pH 8.0, 137 mM NaCl, 1% Triton X-100, 10% glycerol, 2 mM EDTA, and 1× EDTA-free Complete Protease Inhibitor tablet (Roche)] and incubated on ice for 30 min. Extracts were cleared by centrifugation for 10 min at 16,000×*g* at 4 °C. Samples were then treated in 6× SDS protein loading buffer (10% SDS, 1 M Tris–Cl pH 6.8, 50% glycerol, 10% β-mercaptoethanol, 0.03% Bromophenol blue) and subjected to SDS–PAGE. Proteins were then transferred from gels onto PVDF (Amersham) membranes. Anti-FLAG (Sigma, F1804) at 1:3000, anti-V5 (Invitrogen, R960-25) at 1:5000, anti-HA (Sigma, H3663) at 1:3000, anti-GFP (SCBT, sc-9996) at 1:1000, anti-GAPDH (Proteintech, 60004-1-Ig) at 1:3000, and anti-γ-Tubulin (Sigma, T5192) at 1:3000 dilutions were used for immunoblotting analyses. Uncropped western blots are presented in Supplementary Figures 6–8.

**Protein purification of recombinant PSPH and AKR7A2**. Gene-specific primers were used to clone *Bam*HI and *Xho*I restriction endonuclease digestion sites onto the 5′ and 3′ ends, respectively, of ORFs for WT, D32N, T152I, and T149M clones of PSPH by PCR. PCR products as well as a pET28a-based, custom generated pET-6 × His-SUMO expression vector were then digested overnight using *Bam*HI (NEB, R3136) and *Xho*I (NEB, R0146) restriction endonucleases. All digested products were cleaned up by gel extraction. PCR products were then ligated into double-digested pET-6 × His-SUMO vector by 10 μL T4 ligase (NEB, M0202) reactions using a 3:1 ratio of insert to template incubated for 30 min at RT. Ligated products were then transformed into competent cells and plated to isolate single colonies. Properly ligated colonies were validated by colony PCR. Colony PCR-validated pET-6 × His-SUMO PSPH constructs were then transformed into Rosetta strain competent bacteria cells (Novagen, 71401-3).

To purify recombinant WT and mutant PSPH proteins, single colonies of transformed Rosetta strain bacteria were inoculated overnight for use as starter cultures. Starter cultures were used to inoculate 1.0 L LB media including kanamycin and chloramphenicol and incubated for 2–4 h at 37 °C, shaking at 250 rpm until OD600 = 0.6. 200 μL of 1 M IPTG was then added to induce protein expression. Induced cultures were incubated for 18 h at 18 °C, shaking at 250 rpm. After incubation, cultures were centrifuged at 4000×*g* for 20 min at 4 °C.

Supernatant was discarded and pellet was resuspended in 35 mL Resuspension Buffer (500 mM NaCl, 50 mM Tris–base pH 8.0) on ice. Unless stated otherwise, all steps moving forward were performed on ice or at 4 °C. Resuspended pellet was sonicated to lyse cells and then centrifuged at 16,000×$g$ for 45 min. Supernatant was then run through a column prewashed with Wash Buffer (20 mM NaCl, 20 mM Tris pH 7.5) and loaded with Cobalt agarose beads (GoldBio, H-310) for purification of 6× His-tagged protein. Purified samples bound to Cobalt beads were then treated overnight with lab-purified Ulp1 protease for SUMO tag cleavage. Afterwards, samples were again run through a column prewashed with Resuspension Buffer and eluted samples were collected. Lastly, purified protein samples were fractionated by FPLC and samples lacking detectable SUMO expression by Coomassie gel were used for experiments.

WT and mutant A142T recombinant proteins were prepared in the same manner as PSPH except for the following changes: (1) AKR7A2 gene-specific primers were used for PCR, followed by *Eco*RI (NEB, R3101) and *Xho*I (NEB, R0146) double digestion of PCR product and pET-6 × His-SUMO vector; (2) after induction with 200 μL of 1 M IPTG, cultures were incubated for 5 h at 37 °C, shaking at 250 rpm.

**Phosphatase activity measurements for PSPH variants.** WT and mutant PSPH activity were measured using a malachite green assay as follows: Malachite Green Reagent Stock was prepared by combining 30 mL Malachite Green (Sigma, M9636) with 20 mL 4.2% ammonium molybdate (Sigma, 277908)/4 M HCl and mixing for >30 min. Malachite Green Reagent Stock was filtered through a 0.2 μm filter unit and stored at 4 °C. Malachite Green Working Reagent was then prepared by adding Tween-20 to a final concentration of 0.01% in Malachite Green Reagent Stock. Using a 96-well plate (Costar, 3696), $A_{620}$ for sodium phosphate in Malachite Green Working Reagent at concentrations of 10, 15, 20, 25, 30, 35, and 40 μM at pH 7.4 was then measured using a Tecan M1000 plate reader to generate a standard curve. Next, 100 ng of purified recombinant PSPH protein was added to 20 μL total of Assay Buffer (30 mM HEPES at pH 7.4, 1 mM EGTA, 1 mM MgCl$_2$ and 100 μM phosphoserine) and mixed with 80 μL Malachite Green. Negative controls lacking recombinant protein or phosphoserine substrate were also included. After plate incubation at 37 °C for 5 min, $A_{620}$ was measured for all samples. Percent change in phosphatase activity for mutant PSPH was measured as the ratio of mean mutant PSPH activity to mean WT PSPH enzymatic activity over three replicates.

**Kinematic measurement of SSA turnover by AKR7A2.** Using UV-transmitting 96-well plates (Greiner, 675801), $A_{340}$ for NADPH (Cayman, 9000743) at concentrations of 2000, 1000, 500, 250, 125, 62.5, 31.3, 15.6, and 0 μM in 100 μM sodium phosphate buffer at pH 8.0 was measured with a Tecan M1000 plate reader to generate a standard curve. To measure NADPH-dependent turnover of SSA to γ-hydroxybutyrate (GHB) for AKR7A2 WT and mutant A142T, 3.0 μg of purified AKR7A2 protein was added to SSA aliquoted to individual wells in a 96-well plate at concentrations of 0.50, 0.75, 1.0, 1.5, 2.0, 2.5, 3.0, and 3.5 mM in 100 μM sodium phosphate buffer at pH 8.0. Reactions were started simultaneously by adding in NADPH at an initial concentration of 0.5 mM and incubated at 37 °C. Negative controls lacking recombinant protein, NADPH, or SSA were also included. OD$_{320}$ measurements were taken every 60 s for a total of 15 min. AKR7A2 WT and A142T experiments were performed over three replicates.

**Defining duplicate genes and functionally similar proteins.** Duplicate genes were obtained from the Duplicated Genes Database[44], which lists 3543 duplicate genes across 945 different gene groups. In order to compare the robustness of duplicate gene definitions across many different cutoffs, we additionally defined our own metric for protein similarity by running a BLAST of the human proteome against itself and eliminating all pairs of proteins with <40% sequence identity. The remaining pairs were scored using a weighted combination of the pair's percent identity and the coverage with respect to each protein. In Supplementary Fig. 3b, we flexibly defined duplicate genes as all pairs of genes whose score met a minimal duplication threshold tested across all valid ranges (where 0 for Duplication Threshold represents no appreciable similarity and 1 represents perfect identity). Score is calculated as:

$$\text{Score} = \alpha * \text{PercentIdentity} \cdot \text{Coverage}_{\text{Avg}} + (1-\alpha) \cdot \text{Coverage}_{\text{Avg}} \quad (3)$$

where $\alpha=0.95$ and Coverage$_{\text{Avg}}$ is the average coverage between both proteins.

**Enrichment of disruptive mutations on interaction interfaces.** We examined the positions of ExAC variant residues relative to protein–protein interaction interfaces. On interface was defined as either at an interface residue or in the interface domain, while away from interface was defined as neither at an interface residue nor in the interface domain. Solvent accessible surface area calculations were used to define interface residues. Briefly, surface residues whose relative solvent accessibilities change by more than 1 Å$^2$ (between bound and unbound states) are defined to be interface residues. Pfam domains that (1) are known to mediate protein–protein interactions based on 3did[69] or iPfam[70], and (2) contain at least five interface residues are defined as interacting domains. The fraction of interactions disrupted by variants on the interface or away from the interface was then calculated.

**Metrics for evolutionary site conservation at variant sites.** Jensen–Shannon Divergence (JSD) scores were obtained by first performing PSI-BLAST to search for homologs corresponding to proteins tested in this study. The highest scoring homolog by *e*-value per organism is retained, using an *E*-value cutoff of 0.05. A multiple sequence alignment is constructed using the original sequence and retained homolog sequences using Clustal Omega. For all queried proteins with at least 50 homologs, we then calculated JSD scores across all positions in the queried proteins sequence.

phyloP scores were obtained using the Table Browser of the UCSC Genome Browser and inputting the hg19 coordinates for each tested variant. To measure the average global allele frequency across different JSD or phyloP scores, cutoff scores of 0.2, 0.3, …, 1.0 were applied and the global allele frequencies per tested ExAC variant were averaged cumulatively across each cutoff score.

**Signals of positive selection for disruptive alleles.** Fay and Wu's *H* was calculated genome-wide with 1 kb sliding windows using the 1000 Genomes Phase 3 dataset[46]. Analyses were conducted in the merged global population as well as in AFR, EUR, EAS, and SAS populations individually. Genomic regions with a Fay and Wu's *H* statistic at or below the 5th percentile were considered statistically significant. Among all variants that occurred in regions with a measurable Fay and Wu's *H* statistic, the number of disruptive variants that occurred in regions with a significant *H* statistic was recorded.

**Comparing disruption profiles for disease-associated SNVs.** Interaction perturbation data for disease-associated mutations measured here were combined with interaction perturbation data from ref. [19] and then filtered for mutations listed as DM in HGMD (Public release version 2017), resulting in interaction perturbation data for 495 mutations. Mutation pairs were deemed to cause the same disease if strings for their corresponding disease phenotypes listed in HGMD were equal. Mutations were compared pairwise and had to share at least one interaction in common in order to be compared. If one or more interactions was found in common, mutation pairs were categorized by either sharing two or more disrupted interaction in common, one or more disrupted interactions in common, or no disrupted interactions in common.

**Calculating LD between ExAC-tested variants and GWAS SNPs.** To examine whether ExAC variants that are in strong linkage disequilibrium (LD) with GWAS SNPs are more likely to be disruptive, we first extracted all SNPs associated with phenotypes in the UK Biobank GWAS Atlas[71]. We then calculated $R^2$ values between all ExAC variants in our dataset and the UK Biobank GWAS SNPs, using 1000 Genomes Phase 3 data.

ExAC variants in strong LD with GWAS SNPs had a disruption rate that was not significantly different from the overall disruption rate (Supplementary Fig. 5a). Results were robust across multiple $R^2$ thresholds using either African or European allele frequencies (Supplementary Fig. 5a, c). Since most GWAS SNPs occur at MAF ≥ 0.1% and a sizable fraction of our tested variants are rare, we also repeated our analysis restricted for variants at MAF ≥ 0.1% but still found no significant trend (Supplementary Fig. 5b, d). As a control, we also repeated these same analyses using SNPs from the NCBI GWAS Catalog[72] and found the exact same trends as those for the UK Biobank GWAS Atlas (Supplementary Fig. 5e–h).

**Developing a dataset of drug-relevant disruptive SNVs.** We provide background for how our disruptive variant data can be applied to finding drug-relevant disruptive SNVs, including annotation sources for all human enzymes and drug target genes examined, in Supplementary Note 5. Among the SNVs that we tested, 350 were on enzymes, and 84 of them disrupted at least one interaction. More specifically, 54 SNVs were tested on drug-metabolizing enzymes and 12 of them were disruptive. In addition, 227 SNVs were tested on drug targets, 66 of which disrupted at least one interaction. Lastly, five SNVs were tested on drug transporters and three of them were disruptive. As a potential resource to pharmacogenomics and toxicogenomics, we provide a table of all disruptive SNVs that may be relevant to drug action in Supplementary Data 7.

**Generation of mice using CRISPR-Cas9 genome editing.** All animal use was conducted under protocol (2004-0038) to J.C.S. and approved by Cornell University's Institutional Animal Care and Use Committee, affirming that we complied with all relevant ethical requirements for treatment and use of laboratory mice. Optimal guide sequences were evaluated and selected based on high on-target and low off-target scores using Benchling. Previous work has shown that two or more mismatches strongly reduces cutting efficiency, even more so if a PAM site is not present[73]. We note that the closest off-target sequence found with a PAM site had three mismatches and was not found in a gene.

To generate the sgRNA, we used a previously published PCR overlap method[73] described as follows: briefly, overlapping PCR primers, together encoding the T7 promoter, 20-nucleotide guide sequence, and RNA secondary structure sequence, were ordered from IDT. The DNA template was reverse-transcribed using Ambion MEGAshortscript T7 Transcription Kit (cat#AM1354) and resulting sgRNA was purified using Qiagen MinElute columns (cat#28004). For pronuclear injection, the sgRNA (50 ng/μL), ssODN (50 ng/μL, IDT Ultramer Service), and Cas9 mRNA (25

ng/μL, TriLink) were co-injected into zygotes (F1 hybrids between strains FVB/NJ and B6(Cg)-$Tyr^{c-2J}$/J) then transferred into the oviduct of pseudopregnant females. Founders carrying at least one copy of the desired alteration were identified and backcrossed into FVB/NJ. Initial phenotyping was done after one backcross generation and additional phenotyping was done with mice backcrossed at least two or more generations.

While we did not specifically check homozygous $Sept12^{G169E}$ mice for potential off-target edits, we note that in addition to using appropriately selected guides designed to minimize the off-target rate of CRISPR editing in mice, founder mice that contained our desired edit were also bred multiple generations before intercrossing to form homozygous $Sept12^{G169E}$ mice. As such, the probability that an unlinked off-target mutation would happen to persist after backcrossing and then be rendered homozygous only in mice homozygous for the $Sept12^{G169E}$ allele would be extremely low. We[73] and others[74] have extensively sequenced CRISPR-edited mice before using this approach and have found no requirement for off-target sequencing when the expected phenotype matches the introduced edit.

**Genotyping Sept12 mice.** A brief diagram of the CRISPR/Cas9 desired editing site and mice genotyping is provided in Supplementary Fig. 9. For genotyping, we collected toes from 8 to 14-day-old mice and created a crude DNA lysate[75]. PCR, using the following two primers: 5′- GAGATGGGATGACAGGACTATTG-3′ and 5′-GTGGATGAGTGAGGGAAGAAAG-3′, was performed using EconoTaq and associated PCR reagents (Lucigen) with 3 μL of crude DNA lysate. The PCR cycle used for $Sept12^{G169E}$ was: 95 °C for 5 min, 30 cycles of 95 °C for 30 s, 64 °C for 30 s, 72 °C for 30 s, and final elongation at 72 °C for 5 min. To distinguish between WT and G169E, PCR amplicons were digested by $MscI$ to yield WT fragments of 180 and 138 bp, whereas the G169E allele remained uncut.

**Fertility test.** WT, heterozygous, and homozygous males and females were bred to WT counterparts starting at 2 months until 7–15 months of age. The litter size and sex of pups were recorded.

**Sperm motility.** Both epididymides were harvested from adult males, washed in PBS, and placed in a puddle of in vitro fertilization media (Cook Medical). A slit was cut along each epididymis and sperm were allowed to swim out for 2 min at 37 °C. Next, 10 μL of sperm was moved to a glass slide for motility assessment.

**Reporting summary.** Further information on research design is available in the Nature Research Reporting Summary linked to this article.

## Data availability

All mutant clones generated in this study and listed in Supplementary Data 2–4 are available upon request. Please address requests to Haiyuan Yu (haiyuan.yu@cornell.edu). The source data underlying Figs. 1b, 2a, b, d–g, 3c–f, 4a–e, 5b, and f, and Supplementary Figs. 1b–e, 2a, b, 3a, b, 4a, b, and 5a–h are provided as a Source Data file.

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

## Acknowledgements

We thank the members of the Yu Lab for insightful discussions and acknowledge Tommy Vo and Nathaniel Tippens for helpful suggestions. This work was supported by grants from NIGMS (R01 GM097358 to H.Y., R01 GM104424 to H.Y., R01 GM124559 to H.Y., and R01 GM125639 to H.Y. and A.G.C.), NIDDK (R01 DK115398 to H.Y.), NCI (R01 CA167824 to H.Y.), NICHD (R01 HD082568 to J.C.S and H.Y.), NHGRI (UM1 HG009393 to H.Y. and R01 HG006849 to A.G.C. and A.K.), NSF (DBI-1661380 to H.Y.), SFARI (575547 to H.Y.), and contract from NYSTEM (CO29155 to J.C.S.). M.M., P.D.S., and D.N.C. acknowledge receipt of funding from Qiagen Inc. through a License Agreement with Cardiff University.

## Author contributions

H.Y. conceived and oversaw all aspects of the study with input from R.F., J.D., and A.G.C. R.F, J.L, C.A.R.-E., and T.-Y.W. designed and performed experiments. J.D., R.F., S.D.W., S.L., J.F.B., K.Y., and L.Y. performed computational analyses. T.N.T. performed all mice work under the supervision of J.C.S. M.M., P.D.S, D.N.C., and X.W. provided constructive feedback. A.K., A.G.C., and H.Y. supervised research and provided constructive feedback. R.F, J.D., and H.Y. wrote the manuscript with key contributions from S.D.W., J.F.B., A.K., J.C.S., and A.G.C.

## Additional information

**Competing interests:** The authors declare no competing interests.

