## [Peer Review File · Nature Communications]

Reviewers' Comments:

Reviewer #1:

Remarks to the Author:

Report on Fragoza et al "Extensive disruption of protein interactions by genetic variants across the spectrum of allele frequencies in human populations"

This study on variation dependent protein interactions is an excellent piece of work with a quite unexpected outcome. Several lab including Yu et al. have demonstrated that changing interaction pattern are tightly linked to phenotypes and contrast the idea of null vs wt function of a protein. To put simple the functional unit of the cell (which is conceptually most informative) is the interaction rather than the protein. Therefore the effect of SNPs on protein interaction is studied, of course focusing on assaying disease associated variation. Population variation is often used to provide a contrasting picture, a signature of non-functional variation. Fragoza et al. challenge the common view and, using similar high quality pipelines that have been used before to assay disease variation, start testing the effect of population variation on PPI. On the basis of large scale Y2H data the authors test about 2800 amino acid variants in a total of 2800 interactions. The variants are sampled from EXAct data using binned allele frequency. Interaction effects are classified as null (loss of PPI), disruptive and non-disruptive. Importantly they also address empirically the question whether interaction interfaces vs folding through fluorescence screening of 278 ExAC variants. The data are analyses computationally comparing the classes with Grantham scores, a biochemical measure quantifying the dissimilarity between amino acid residues, Polyphen2 predicitions, according to interaction surface etc. They also show that disruptive variants show positive selection signature when compared to often non-disruptive variants. The analysis suggest that results of population variation SNPs are not that different from results obtained with disease association SNPs.

The surprising outcome is therefore that a substantial fraction of non-disease associated amino acid changes shows an interaction perturbation effect. This includes a fraction of common alleles thus implying that the interaction is perturbed in a large number of people. The hypothesis is that these (common) alterations will provide a background ("epistatic relationships") for rendering these individuals susceptible to non-highly penetrant disease variation (that e.g. effects interactions in the network neighbourhood).

Points for consideration:

*) The study shows quite some breath with regards to the variants tested. The outcome establishes that essentially testing variation in great depth would be very important. The authors should therefore contrast/discuss their approach with/in the context of the scanning mutagenesis PPI approaches from other labs (i.e. Fields/Shendure, Stelzl, Lehner), which provide required depth to study >20 Mio SNVs.

*) I am slightly confused about the validation and the examples picked. In both cases, PSPH and AKR7A2, enzymatic activity of disrupting non-disease mutations are tested, i.e. node function. The authors do not properly establish the context of the interaction profile and the activity changes. I assume it is sort of trivial in that both proteins function as homo-dimer - and one of the interaction losses is the self-interaction. Are this prominent cases? How to reconcile this with the general idea of protein interaction perturbation as disease marker etc...

*) I think the authors could do a slightly better job in presenting the actual perturbation data. X number of interactions against y number of alleles can mean many things, also that only a very few alleles have an effect on very many interactions. Though there is a graph that tells about the allele distribution of the targeted alleles, it remains unclear what the outcome in terms of PPI network is. Again, e.g. ... 445 Disruptive SNVs and 4,761 SNV-Interaction Pairs ..., how do the distributions look like?

Reviewer #2:

Remarks to the Author:

This is a remarkable study providing evidence for functionality (protein-protein interactions, PPI) of missense protein variants at about twice the level previously claimed, and notably including an unexpectedly large proportion, around 10%, of common non-synonymous variants in the human genome. The study combines high throughput protein mutagenesis and cellular/biochemical assays with useful bioinformatics. I do not have any major issues with the manuscript, but suggest one further analysis that will address the possible phenotypic consequences of the PPI, and a couple of items of clarification.

While the study is well motivated and generally well described, I would like to see more of a description of how the protein partners for the interaction screen were chosen. It appears that the choice of 2008 missense variants was somewhat random, ensuring coverage of all frequency classes, but then how does this lead to 2181 PPI tests? It is not clear to me from the methods, not being a Y2H practitioner, whether it was an all-against-all comparison (namely ~4 million pairwise contrasts of mutant against wt of which 2181 involve clear wt-wt PPI) or was some procedure used to reduce the search space? Were the tests calibrated against a known databases of interactions? In any case, what is the likely impact on the false-positive and false-negative rates of discovery? How many of the potential partners of each mutant protein have been screened, and does this matter?

Similarly, with regard the Protein Complementation Assay, please provide some background in the text regarding what it is testing. Again, the methods are rather technical and inaccessible for non-experts.

The authors have performed an impressive series of analyses designed to evaluate the impacts of allele frequency, cancer driver/HGMD, protein structure and stability, and evolutionary conservation. The one additional analysis that I would request would be to evaluate whether there is a bias for the PPI-disrupting variants to be more likely to be associated with phenotypes by GWAS. This has very recently become possible through the publication of the UK Biobank GWAS Atlas by Albert Tenesa's group (<http://geneatlas.roslin.ed.ac.uk>) Nature Genetics 50, 1093-8, Nov 2018), which documents associations for 10 million variants with 780 binary and continuous traits. I recognize that a thorough analysis including fine-mapping LD adjustment would be a lot of work, but an initial scan should definitely be feasible.

Reviewer #3:

Remarks to the Author:

In this manuscript, the authors describe their study on mutations disrupting protein-protein interactions (PPIs) among human genome SNVs.

They extracted SNV data from ExAC dataset and performed large scale yeast two-hybrid assays to evaluate whether these mutations affect these selected PPIs. Several interesting observations were obtained:

- (i) PPI disruptive SNVs are prevalent
- (ii) Their fraction is reversely proportional to MAF
- (iii) They are more enriched in disease-associated genes.

Although the biological significance and medical inference of disruptive SNVs is important, the authors do not provide any serious biological drill downs that would confirm their PPI data - they definitely need to address this issue by providing such biological drill downs on one or two selected PPI pairs. As a suggestion, the synthetic lethality, either genetically or pharmacologically, may help them to validate some of their results.

In consistency with the above, the PSPH T152I SNV needs further exploration. It exhibits the same enzymatic characteristics as the disease-associated mutation D32N. Why it does not cause any

pathological phenotype as D32N mutant should be explained.

The authors provide two examples of functional defects of SNVs. These were demonstrated by in vitro enzymatical assays which do not require any PPI in the reactions. It is necessary to distinguish the deleterious effects on PPIs and on enzyme activities, which should not be the same.

Furthermore, the estimation of many parameters was based on two PPI assays: YTH and PCA. As the data provided by the authors and work from many previous studies, the recovery rate is pretty low. This might skew the estimation of the true value of disruptive SNV fraction, and should definitely be notified.

In addition, the authors mention that a disruption is mild and only partially interferes the involved biochemical process. I think it's also possible that the disruption can be compensated by redundant parallel pathways.

Moreover, the authors did not describe the whole scenario of the medical inference. They suggested that a SNV might contribute liability to certain genetic diseases when combined with mutations in other genes involved in the same biochemical process. However, this is a rare situation.

I believe the results of this study might have more implications in pharmacogenomics and toxicogenomics. For example, the disruptive SNVs may affect the sensitivities to certain drugs and environmental cues.

Point-by-Point Response Letter

Ref1.1 – “an excellent piece of work with a quite unexpected outcome.” –

Reviewer Comment	This study on variation dependent protein interactions is an excellent piece of work with a quite unexpected outcome. Several lab including Yu et al. have demonstrated that changing interaction pattern are tightly linked to phenotypes and contrast the idea of null vs wt function of a protein. To put simple the functional unit of the cell (which is conceptually most informative) is the interaction rather than the protein. Therefore the effect of SNPs on protein interaction is studied, of course focusing on assaying disease associated variation. Population variation is often used to provide a contrasting picture, a signature of non-functional variation. Fragoza et al. challenge the common view and, using similar high quality pipelines that have been used before to assay disease variation, start testing the effect of population variation on PPI. On the basis of large scale Y2H data the authors test about 2800 amino acid variants in a total of 2800 interactions. The variants are sampled from EXAct data using binned allele frequency. Interaction effects are classified as null (loss of PPI), disruptive and non-disruptive. Importantly they also address empirically the question whether interaction interfaces vs folding through fluorescence screening of 278 ExAC variants.
Author Response	We thank the reviewer for their comments. We appreciate that the reviewer recognizes the significance of challenging the common view that most population variation is non-functional.

Ref1.2 – “a substantial fraction of non-disease associated amino acid changes shows an interaction perturbation effect.” –

Reviewer Comment	The data are analyses computationally comparing the classes with Grantham scores, a biochemical measure quantifying the dissimilarity between amino acid residues, Polyphen2 predicitions, according to interaction surface etc. They also show that disruptive variants show positive selection signature when compared to often non-disruptive variants. The analysis suggest that results of population variation SNPs are not that different from results obtained with disease association SNPs. The surprising outcome is therefore that a substantial fraction of non-disease associated amino acid changes shows an interaction perturbation effect. This includes a fraction of common alleles thus implying that the interaction is perturbed in a large number of people. The hypothesis is that these (common) alterations will provide a background (“epistatic relationships”) for rendering these individuals
------------------	---

	susceptible to non-highly penetrant disease variation (that e.g. effects interactions in the network neighbourhood).
Author Response	We thank the reviewer for confirming that our finding that a large fraction of SNPs disrupt protein interactions is indeed novel. We further agree that our results imply that disruptive, common alleles would cause corresponding interaction perturbations in a large number of people, and that these perturbations provide the genetic background for rendering individuals' susceptibility to many disease mutations that are not highly penetrant.

Ref1.3 – Our mutagenesis approach complements and is orthogonal to previous mutagenesis studies –

Reviewer Comment	*) The study shows quite some breath with regards to the variants tested. The outcome establishes that essentially testing variation in great depth would be very important. The authors should therefore contrast/discuss their approach with/in the context of the scanning mutagenesis PPI approaches from other labs (i.e. Fields/Shendure, Stelzl, Lehner), which provide required depth to study >20 Mio SNVs.
Author Response	We thank the reviewer for noting that testing variation in great depth is very important. Indeed, previous studies have also aimed to experimentally measure the impact of coding mutations at large scales through contrasting approaches. For example, Fields and Shendure developed a massively parallel single-amino-acid mutagenesis pipeline named PALS that can generate nearly all potential singleton mutations possible for a particular gene of interest (Kitzman et al. Nature Methods 2015). This impressive depth makes PALS an excellent method for studying extensive variation in a single protein; however, PALS is not optimized for studying variants of interest across multiple different genes. Our mutagenesis approach allowed us to study >2,000 mutations across 847 unique genes. In contrast, PALS has only been applied to a handful of genes, notably TP53 (Kitzman et al. Nature Methods 2015) and BRCA1 (Findlay et al. Nature 2018). Regardless of whether extensive mutations are generated for a single gene or across a diverse set of genes, the impact of a mutation must be characterized through functional assays. Y2H is widely used for characterizing the impact of mutations on protein-protein interactions (Wei et al. PLOS Genet 2014; Sahni et al. Cell 2015), but several derivatives for detecting perturbations by Y2H exist. For example, Stelzl and colleagues developed the Int-Seq platform for probing protein-protein interaction disruptions using a Reverse Two-Hybrid (R2H) approach (Woodsmith et al. Nature Methods 2017). Specifically, a two reporter system is used such

	that positive yeast growth occurs when an interaction is perturbed. While this R2H approach increases assay sensitivity for disruptive variants, a R2H reference interactome of known wild-type interacting protein pairs does not exist yet. Therefore interacting wild-type protein interaction pairs must be screened first before corresponding disruptive variants can be subsequently identified, which can limit the throughput of this method. Indeed, Stelzl and colleagues limited their screen to disruptive mutations on eight proteins across nine total interactions (Woodsmith et al. Nature Methods 2017). In contrast, a reference interactome of >14,000 interaction pairs specific to our form of Y2H was already available which allowed us directly screen the impact of our >2,000 SNVs across 2,185 unique protein-protein interactions. Mutagenesis pipelines can be combined with numerous different functional assays beyond protein interactions. For example, Lehner and colleagues used the deep mutational scanning pipeline developed by Fowler and Fields (Fowler and Fields. Nature Methods 2014) to examine the impact of mutations on alternative splicing of the FAS/CD95 exon 6, which is involved in the control of cellular apoptosis (Julien et al. Nat Commun 2016). Notably, in addition to examining nearly all single mutations in the 63-nucleotides-long FAS/CD95 exon 6, Lehner and colleagues also examined how mutation pairs impact splicing of this exon. Examining pairs of mutations allowed the authors to dissect how epistatic relationships between variants impact alternative splicing, a unique strength to their study; however, this epistatic examination was limited to a single exon, as opposed to examining mutational effects across a diverse breadth of genes as performed in our study. We have updated the discussion section of our manuscript with this comparative methods analysis discussed here.
Excerpt from Revised Manuscript	[Page 12] Several methods to experimentally measure the impact of coding mutations at large scales have been recently reported^{20,74-76}. The depth of proteins, variants, and interactions presented here complements these previous methods well. For example, Fields and Shendure developed a massively parallel single-amino-acid mutagenesis pipeline, named PALS, that can generate nearly all potential singleton mutations possible for a particular gene of interest⁷⁴. This impressive depth makes PALS an excellent method for studying extensive variation in a single protein, most notably TP53⁷⁴ and BRCA1⁷⁷ but remains to be optimized for studying variation across a large set of unique genes. In contrast, our mutagenesis approach allowed us to survey >2,000 mutations across 847 unique genes. Similarly, while Y2H is widely used for characterizing the impact of mutations on protein-protein interactions^{19,20}, several derivatives for detecting perturbations by Y2H exist. For instance, Stelzl and colleagues developed the Int-Seq platform for probing protein-protein interaction disruptions using a Reverse Two-Hybrid (R2H) approach⁷⁶. While this R2H approach increases assay sensitivity, a R2H reference interactome is not yet available, limiting the coverage of this approach to a handful of interactions. [Page 13]

	For example, Lehner and colleagues a deep mutational scanning pipeline ⁷⁸ to measure the impact of mutations on alternative splicing ⁷⁹ . Continued efforts to survey all potential manners in which molecular-level perturbations can alter cellular and organismal phenotypes are needed to properly understand the impact of mutations on human health.
--	--

Ref1.4 – Better establishing the context of protein interaction perturbations as disease markers –

Reviewer Comment	*) I am slightly confused about the validation and the examples picked. In both cases, PSPH and AKR7A2, enzymatic activity of disrupting non-disease mutations are tested, i.e. node function. The authors do not properly establish the context of the interaction profile and the activity changes. I assume it is sort of trivial in that both proteins function as homo-dimer - and one of the interaction losses is the self-interaction. Are this prominent cases? How to reconcile this with the general idea of protein interaction perturbation as disease marker etc...
Author Response	We appreciate the opportunity to better clarify how the interaction profiles for our enzymatic examples relate to changes in protein activity. We also use this opportunity to (1) introduce an example of two disease mutations on SMAD4 with matching disruption profiles that result in the same disease and a third non-disruptive SMAD4 mutation that results in a clinically distinct disease; (2) clarify how a disruption to PSPH and AKR7A2 dimerization can also impact enzymatic activity and comment on how prominent these cases may be; and (3) introduce an additional functional study in which we generated CRISPR-edited mice homozygous for a disruptive rare variant in SEPT12 and showed that this mutation resulted in male subfertility. We first observed that pairs of disease-associated mutations on the same gene that disrupt the same set of interactions result in the same disease significantly more often than mutation pairs with differing interaction perturbation profiles (Fig. 4e). To complement this observation, we have highlighted three disease-associated mutations on the protein SMAD4 (Fig. 4f), a crucial protein in the TGFβ/SMAD signaling pathway. Two mutations on SMAD4, E330K and G352R, result in juvenile polyposis (Gallione et al. Am J Med Genet 2010; Sayed et al. Ann Surg Oncol 2002) while a third mutation, N13S, results in a clinically distinct disease, pulmonary arterial hypertension (Nasim et al. Hum Mut 2011). We observed that E330K and G352R cluster together in three dimensional space near the SMAD4-SMAD3 interaction interface (Fig. 4g). N13S, in contrast, is positioned away from E330K and G352R near the N-terminus of SMAD4. In agreement with the proximal clustering of E330K and G352R near the SMAD4-SMAD3 interaction interface, both mutations disrupted the SMAD4

interaction with SMAD3 in addition to disrupting the SMAD4-SMAD9 interaction (**Fig. 4f**). These SMAD protein disruption results agree with previous evidence implicating the TGF β /SMAD signaling pathway in the formation of juvenile polyposis (Jung et al. *Gastroent* 2017; Massangué. *Cell* 2008). In contrast, the N13S mutation left SMAD4 interactions with SMAD3 and SMAD9 intact, which agrees with a previous study that found no evidence that N13S alters SMAD-mediated signaling (Nasim et al. *Hum Mut* 2011). Collectively, this result demonstrates how mutations with the same disruption profiles can lead to the same disease phenotype. With this result as a template, we then proceeded to explore cases in which both a SNV and a known disease-associated mutation shared the same disruption profile with the goal of determining whether the disruptive SNV also resulted in the same disease phenotype.

In our disruption assays, we observed that a PSPH disease-associated mutation, D32N, as well as a rare variant, T152I, both disrupted interactions with itself while leaving all other PSPH interactions intact (**Fig. 5a**). Previous research has reported the importance of PSPH dimerization, noting that PSPH is a dimer in solution and that “mutant residues such as Y138K, F139K, and Y143K, which could interfere with dimer interfaces, exist in an aggregated and insoluble form” (Kim H-Y et al. *JBC* 2002). Since the T152I rare variant disrupts an interaction with itself, we hypothesized that T152I would also reduce enzymatic activity. Indeed, *in vitro* measurements of PSPH enzymatic activity showed that the T152I mutant protein reduced activity to the same extent as D32N while a non-disruptive mutation, T149M, left enzymatic activity intact (**Fig. 5b**). Lastly, we note that dimerization may also be crucial for AKR7A2 enzymatic activity since the mutation A142T both disrupts an interaction with itself (**Fig. 3f**) and reduces AKR7A2 enzymatic activity (**Fig. 3g**).

While a shared PSPH self-disruption between the rare variant T152I and D32N allowed us to prioritize T152I as a candidate disease-associated mutation, only 61 out of 669 (9.1%) total disrupted interactions in our dataset involved protein homodimers. In an effort to show that shared disruption profiles can be applied to other categories of interaction perturbations, we also characterized disruptive mutations on the GTPase, SEPT12.

SEPT12 is known to interact with other septin proteins found in the SEPT2, SEPT6, and SEPT7 protein subgroups, to form a filamentous structure at the sperm annulus (Mostowy and Cossart *Nat Rev Mol Cell Biol* 2012; Sellin et al. *Mol Biol Cell* 2014; Kuo et

	al. J Cell Sci 2015). Mutation-induced perturbations to SEPT12 interactions with these septin subgroup proteins can compromise the structural integrity of the sperm annulus to result in male infertility. Indeed, an infertility-causing mutation in men, SEPT12_D197N, was previously shown to disrupt interactions with septin proteins found at the sperm annulus, resulting in a disorganized sperm annulus and poor sperm motility in a mouse model for SEPT12_D197N (Kuo et al. J Cell Sci 2015). We identified a rare variant in SEPT12, G169E (MAF = 0.02%), that disrupted interactions with SEPT7 as well as the SEPT2 subgroup proteins, SEPT1 and SEPT5 (Fig. 5c). The infertility-causing mutation, D197N, disrupted these exact same septin interaction partners by Y2H. Using homology modeling, we observed that both G169E and D197N mutations occur at SEPT12 interaction interface residues with SEPT1 (Fig. 5d) and confirmed that both mutations disrupt the SEPT12-SEPT1 interaction without reducing protein stability in 293T cells (Fig. 5e). These results demonstrate that these mutations function by specifically perturbing SEPT12 protein-protein interactions as opposed to disrupting SEPT12 stability as a whole. To investigate whether these matching SEPT12 molecular phenotypes result in corresponding organismal phenotypes, we generated Sept12^{G169E} mice using a CRISPR-editing approach. We found that homozygous Sept12^{G169E} males were subfertile in comparison to wild-type males (Fig. 5f). Notably, sperm from homozygous Sept12^{G169E} males exhibited poor motility (Fig. 5g), a phenotype also reported for Sept12^{D197N} male mice (Kuo et al. J Cell Sci 2015). Taken in context with our in vitro disruption data, our mouse model strongly suggests that SEPT12_G169E is an infertility-associated mutation in men and further demonstrates how shared disruption profiles can be used to prioritize candidate disease-associated mutations. Lastly, we note that no individuals homozygous for SEPT12_G169E are reported in ExAC, potentially explaining why SEPT12_G169E has not been previously reported as an infertility-associated variant. The manuscript has been updated to include our SMAD4 disruption profile examples, better establish the context of our PSPH and AKR7A2 disruption profiles, and to include our SEPT12 subfertility example.
Excerpt from Revised Manuscript	[Page 7] Since AKR7A2 is a dimer in solution and A142T disrupts an AKR7A2 interaction with itself, we hypothesized that this mutation might also impact AKR7A2 enzymatic activity. As such, we

purified recombinant wild-type and mutant AKR7A2 protein to test for changes in NADPH-dependent turnover of SSA (**Methods**).

[Page 9]

To demonstrate how pairs of disease-associated mutations on the same gene with matching disruption profiles can result in the same disease, we highlight three disease-associated mutations on SMAD4 (**Fig. 4f**), a crucial protein in the TGF β /SMAD signaling pathway. Two mutations on SMAD4, E330K and G352R, are associated with juvenile polyposis^{54,55} while a third mutation, N13S, results in a clinically distinct disease, pulmonary arterial hypertension⁵⁶. We observed that E330K and G352R cluster together in three dimensional space near the SMAD4-SMAD3 interaction interface (**Fig. 4g**). N13S, in contrast, appears positioned away from E330K and G352R near the N-terminus of SMAD4. In agreement with the proximal clustering of E330K and G352R near the SMAD4-SMAD3 interaction interface, both mutations disrupted the SMAD4 interaction with SMAD3 in addition to disrupting the SMAD4-SMAD9 interaction (**Fig. 4f**). These SMAD protein disruption results agree with previous evidence implicating the TGF β /SMAD signaling pathway in the formation of juvenile polyposis^{57,58}. In contrast, the N13S mutation left SMAD4 interactions with SMAD3 and SMAD9 intact, which agrees with a previous study that found no evidence that N13S alters SMAD-mediated signaling⁵⁶.

With this example as a template, we then explored cases in which both an ExAC variant and a known disease-associated mutation shared the same disruption profile with the goal of determining whether the population variant exhibited evidence of the same disease phenotype. To do this, we tested two mutations with matching disruption profiles on the protein PSPH (**Fig. 5a**): (i) T152I, a rare variant (MAF = 0.10%) in ExAC that disrupts an interaction with itself and (ii) D32N, which also disrupts an interaction with itself and causes phosphoserine phosphatase deficiency in a compound heterozygous individual with two deleterious PSPH mutations⁵⁹. An additional PSPH non-disruptive rare variant, T149M, was included as a control. Since PSPH exists as a dimer in solution and can aggregate when mutations that interfere with dimerization are introduced⁶⁰, we reasoned that mutations that disrupt this dimerization may also reduce PSPH enzymatic activity. We therefore purified recombinant wild-type, D32N, T152I, and T149M PSPH proteins and measured for changes in phosphatase activity for PSPH mutants relative to wild-type using a malachite green assay.

[Page 10]

To further show how potentially physiologically-relevant mutations can be identified using shared disruption profiles, we also characterized a pair of disruptive mutations on the GTPase, SEPT12: a rare variant not known to associate with any disease phenotypes, G169E (MAF = 0.02%), and D197N, an infertility-causing mutation in men⁶¹. Both mutations perturbed interactions with SEPT7 and SEPT2 subgroup proteins, SEPT1 and SEPT5 (**Fig. 5c**). These perturbations are particularly relevant because SEPT12 is known to interact with other septin proteins found in the SEPT2, SEPT6, and SEPT7 protein subgroups to form a filamentous structure at the sperm annulus⁶²⁻⁶⁴. Moreover, the infertility-causing mutation SEPT12_D197N, which was previously shown to perturb interactions with these same septin subgroup proteins, resulted in a disorganized sperm annulus and poor sperm motility in a mouse model for D197N⁶⁴. Lastly, using homology modeling, we observed that both G169E and D197N mutations occur at SEPT12 interaction interface residues with SEPT1 (**Fig. 5d**) and confirmed that both mutations disrupt the SEPT12-SEPT1 interaction without reducing protein stability in 293T cells (**Fig. 5e**). These results demonstrate that these mutations function by specifically perturbing SEPT12 protein-protein interactions as opposed to disrupting SEPT12 stability as a whole.

We then investigated whether these matching SEPT12 molecular phenotypes result in corresponding organismal phenotypes by generating *Sept12*^{G169E} mice using a CRISPR-editing approach⁶⁵. We found that homozygous *Sept12*^{G169E} males were subfertile in comparison to wild-type males (**Fig. 5f**). Notably, sperm from homozygous *Sept12*^{G169E} males exhibited poor motility (**Fig. 5g**), a phenotype also reported for *Sept12*^{D197N} male mice⁶⁴. These observations of poor sperm motility and subfertility in mice suggest that SEPT12_G169E may deleteriously impact fertility in men homozygous for this mutation, although we note that no individuals homozygous for SEPT12_G169E are reported in ExAC. Taken together with our *in vitro* data, these results also demonstrate how shared disruption profiles can be used to prioritize candidate disease-associated mutations.

Ref1.5 – Clearer presentation of SNV perturbation data –

Reviewer Comment	*) I think the authors could do a slightly better job in presenting the actual perturbation data. X number of interactions against y number of alleles can mean many things, also that only a very few alleles have an effect on very many interactions. Though there is a graph that tells about the allele distribution of the targeted alleles, it remains unclear what the outcome in terms of PPI network is. Again, e.g. ... 445 Disruptive SNVs and 4,761 SNV-Interaction Pairs ..., how do the distributions look like?
Author Response	In an effort to present our perturbation data more transparently, we first plotted a pie chart categorizing the number of variants that disrupt either 1, 2, 3, 4, or 5 or more interactions (Supplementary Fig. 1c). We observed that 205 of our tested SNVs disrupt only a single interaction (68.8%), suggesting that disruptive mutations perturb specific subsets of protein function as opposed to perturbing protein function as a whole. Our pie chart also revealed that many of the SNVs that we tested disrupted five or more interactions (6.7%). To further explore the extent to which highly disruptive SNVs specifically contribute to the total number of disrupted interactions reported in our data, we constructed a cumulative distribution function (CDF) plotting the fraction of disruptive SNVs against the proportion of total disrupted interactions (Supplementary Fig. 1d). Notably, we found that four outlier SNVs contributed to 26.6% of the total disrupted interactions reported in our study, a moderate impact. We note, however, that the analyses in our study are largely calculated from the fraction of disruptive variants (i.e. variants that disrupt ≥ 1 interaction are scored as disruptive) as opposed to the fraction of disrupted interactions, and therefore any skew resulting from this handful of highly disruptive variants does not affect our conclusions. Nonetheless, we re-examined the only per-interaction analysis in our manuscript that includes interactions from these outliers: our PCA retest of disrupted and non-disrupted SNV-interactions (Fig. 2a). Repeating this PCA retest calculation with any interactions corresponding to these four outlier SNVs removed showed no changes in our results (Supplementary Fig. 1b). The manuscript has been updated to include the analyses and discussion presented here.
Excerpt from Revised Manuscript	[Page 4] We further validated the quality of our SNV-interaction network by performing Protein Complementation Assay (PCA)⁴⁰ in human 293T cells to retest a representative subset of ~400 disrupted and non-disrupted SNV-interactions pairs amongst our ExAC subset. SNV-disrupted interactions retested at a rate approximate to a negative reference set comprising randomly selected ORF pairs whereas non-disrupted interactions retested at a rate statistically

	indistinguishable from a positive reference set of literature-established protein interactions^{41,42} (Fig. 2a, Supplementary Fig. 1a). Our result remained unchanged when we removed interactions corresponding to highly-disruptive SNVs (Supplementary Fig. 1b). [Page 5] The extent to which a mutation is disruptive can also be categorized by measuring the fraction of corresponding protein interactions disrupted by a particular variant. Accordingly, we first grouped each of our 298 disruptive variants by the number of interactions they perturb (Supplementary Fig. 1c). We observed that 205 of our tested SNVs disrupted only a single interaction (68.8%) while a small fraction of variants (6.7%) disrupted five or more interactions, suggesting that disruptive mutations tend to perturb specific subsets of protein function as opposed to perturbing protein function as a whole. Examining the distribution of disruptive variants across the number of interactions perturbed revealed a similar trend (Supplementary Fig. 1d).
--	---

Ref2.1 – “a remarkable study providing evidence for functionality of missense protein variants.” –

Reviewer Comment	This is a remarkable study providing evidence for functionality (protein-protein interactions, PPI) of missense protein variants at about twice the level previously claimed, and notably including an unexpectedly large proportion, around 10%, of common non-synonymous variants in the human genome. The study combines high throughput protein mutagenesis and cellular/biochemical assays with useful bioinformatics. I do not have any major issues with the manuscript, but suggest one further analysis that will address the possible phenotypic consequences of the PPI, and a couple of items of clarification.
Author Response	We thank the reviewer for confirming the validity and importance of our study.

Ref2.2 – Unbiased protein interaction perturbation screens –

Reviewer Comment	While the study is well motivated and generally well described, I would like to see more of a description of how the protein partners for the interaction screen were chosen. It appears that the choice of 2008 missense variants was somewhat random, ensuring coverage of all frequency classes, but then how does this lead to 2181 PPI tests? It is not clear to me from the methods, not being a Y2H practitioner, whether it was an all-against-all comparison (namely ~4 million pairwise contrasts of mutant against wt of which 2181 involve clear wt-wt PPI) or was some procedure used to reduce the search space? Were the tests calibrated against a known databases of interactions? In any case, what is the likely impact on the false-positive and false-negative rates of discovery? How many of the potential partners of each mutant protein have been screened, and does this matter?
Author Response	To select interaction partners for each mutant protein tested in our SNV-perturbation screen, we first leveraged a Y2H reference interactome comprised of over 14,000 known wild-type protein-protein interactions reported in four manuscripts (Rual et al. Nature

2005, Venkatesan et al. *Nature Methods* 2009, Yu et al. *Nature Methods* 2011, Rolland et al. *Cell* 2014). Because these are published interactions that are retestable by our version of Y2H, we did not perform an all-by-all interaction screen which would require testing all potential interaction partners for a particular protein (in other words, we did not perform ~4 million pairwise tests); rather, we only retested interactions already reported in this reference interactome. Only interactions from this wild-type reference interactome that retest positive in our lab are considered in our analyses. This requirement for retestable wild-type interactions found in the literature-reported reference interactome dramatically reduces the search space in which we probe for disruptive SNVs.

On average, each protein in the reference interactome has between 2-3 interaction partners. We note that we tested 847 unique genes against 2,185 corresponding interaction partners (~2.5 interaction partners per gene-encoded protein). For each wild-type protein-protein interaction, we then tested whether corresponding SNVs for each interaction can perturb that interaction. This consisted of 2,009 SNVs found on 847 unique genes. Since each of these proteins has ~2.5 interaction partners, this resulted in a total of 4,797 SNV-interaction pairs tested, as diagrammed in Fig. 1c.

False positive and false negatives must be carefully considered in any high-throughput assay. First of all, our version of Y2H has been applied to screen *hundreds of millions of protein pairs* in five different organisms (Walhout et al. *Science* 2000; Boulton et al. *Science* 2002; Rual et al. *Nature* 2005; Yu et al. *Science* 2008; Arabidopsis Interactome Mapping Consortium. *Science* 2011; Rolland et al. *Cell* 2014; Vo et al. *Cell* 2016) and has been consistently shown to be of high quality, as confirmed by other groups (Lim et al. *Cell* 2006; Tonikian et al. *PLOS Biol* 2009; Bandyopadhyay et al. *Nature Methods* 2010; Kahle et al. *Hum Mol Genet* 2011; Soler-Lopez et al. *Genome Research* 2011). The false positive rate has also been consistently shown to be minimal (measured as <1% in human, Braun et al. *Nature Methods* 2009). Thus, the high quality Y2H datasets on human protein-protein interactions provide an excellent starting point for our study.

As with any assay, Y2H does have false negatives (i.e., true interactions not detected by Y2H). However, these false negatives have no bearing on the disruptions we detect, because we test the effect of mutations only on those wild-type interactions that can be detected by our Y2H assay. Thus, the number of disruptions that

	we detect and the corresponding fraction of disruptive SNVs presented in the manuscript are lower bound estimates of the actual number of disruptive SNVs. If we could detect additional interactions (i.e., have an assay with a lower false negative rate), the extent of disruptions detected would only be higher, further highlighting the significance of our findings. Specific for our study, we took systematic steps to ensure the highest possible data quality. First, as stated earlier, we only examined previously reported interactions found in the reference interactome. Second, we required that all reported SNV-disrupted interactions reproduce themselves across three independent screens. Furthermore, the corresponding wild-type interaction is retested in each of these three screens to ensure that the change in Y2H growth phenotype between wild-type and mutant is consistent. While our requirement for mutant interaction pairs to reproduce across three screens minimizes false positives (i.e. a false disruption is very unlikely to reproduce three separate times), this stringency could potentially increase the occurrences of false negatives (i.e. a true disruptive variant may fail to retest across one of three screens). In designing our experiments, we expected that a majority of population variants would be non-disruptive, so we designed our assays to maximize our confidence in the SNVs that we call disruptive, which required a more conservative scoring approach as a result. We have amended our methods section with additional information regarding how interaction partners were selected for testing.
Excerpt from Revised Manuscript	[Supplementary Note 1] To select interaction partners for each mutant protein tested in our SNV-perturbation screen, we first leveraged a Y2H reference interactome comprised of over 14,000 known wild-type protein-protein interactions reported in four manuscripts¹⁻⁴. Since these published interactions are retestable by our version of Y2H, we retested only interactions corresponding to our SNVs of interest that were already reported from this reference interactome. Only interactions from this wild-type reference interactome that retested positive in our lab are considered in our analyses. This requirement for retestable wild-type interactions found in the literature-reported reference interactome dramatically reduces the search space in which we probe for disruptive SNVs and prevented the need for an all-by-all Y2H interaction screen. On average, each protein in the reference interactome has between 2-3 interaction partners. We note that we tested 847 unique genes against 2,185 corresponding interaction partners (~2.5 interaction partners per gene-encoded protein). For each wild-type protein-protein interaction, we then tested whether corresponding SNVs for each interaction can perturb that interaction. This consisted of 2,009 SNVs found on 847 unique genes. Since each of these genes has ~2.5 interaction partners, this results in a total of 4,797 SNV-interaction pairs tested (Fig. 1c).

Ref2.3 – Background information for Protein Complementation Assay –

Reviewer Comment	Similarly, with regard the Protein Complementation Assay, please provide some background in the text regarding what
------------------	---

	it is testing. Again, the methods are rather technical and inaccessible for non-experts.
Author Response	We thank the reviewer for their suggestion. Protein Complementation Assay (PCA) is a protein-protein interaction assay performed in HEK 293T cells in which a bait and prey protein are fused to two complementary fragments of a fluorescent protein, YFP. If the bait and prey protein successfully interact, the two YFP fragments will stably bind and fluoresce as a result. PCA is a particularly valuable assay in protein-protein interaction screens because it is high-throughput, and it provides an independent assay for validating the quality of protein-protein interactions detected through Y2H screens. For this reason, PCA is commonly used in many interactome screens, including in Arabidopsis (Arabidopsis Interactome Mapping Consortium. Science 2011), yeast (Yu et al. Science 2008; Vo et al. Cell 2016), and human (Rolland et al. Cell 2014). Notably, PCA can also be used to validate that two proteins do not interact, which is important when testing the impact of disruptive variants. To do this, we measure the loss of fluorescence signal in PCA for mutant interaction pairs relative to wild-type pairs to validate that Y2H-tested mutations are indeed disruptive (Sahni et al. Cell 2015; Yi et al. Nat Protoc 2017). We have updated our manuscript to provide more background on how PCA functions and how it is used to validate disrupted interactions.
Excerpt from Revised Manuscript	[Supplementary Note 5] Protein Complementation Assay (PCA) is a protein-protein interaction assay performed in HEK 293T cells in which a bait and prey protein are fused to two complementary fragments of a fluorescent protein, YFP. If the bait and prey protein successfully interact, the two YFP fragments will stably bind and fluoresce as a result. PCA is a particularly valuable assay in protein-protein interaction screens because it is high-throughput, and it provides an independent assay for validating the quality of protein-protein interactions detected through Y2H screens. For this reason, PCA is commonly used in many interactome screens, including in Arabidopsis⁷, yeast^{3,8}, and human^{4,14}. Importantly, PCA can also be used to validate that two proteins do not interact, which is insightful when testing the impact of disruptive variants. To do this, loss of fluorescence signal in PCA for mutant interaction pairs relative to wild-type pairs is measured to validate that Y2H-tested mutations are indeed disruptive^{15,16}.

Ref2.4 – PPI-disrupting variants are not biased towards GWAS phenotypes –

Reviewer Comment	The authors have performed an impressive series of analyses designed to evaluate the impacts of allele frequency, cancer driver/HGMD, protein structure and stability, and evolutionary conservation. The one additional analysis that I would request would be to evaluate whether there is a bias for the PPI-disrupting variants to be more likely to be associated with phenotypes by GWAS. This has very recently become possible through the publication of the UK Biobank GWAS Atlas by Albert Tenesa's group (http://geneatlas.roslin.ed.ac.uk) Nature Genetics 50,
------------------	--

	1093-8, Nov 2018), which documents associations for 10 million variants with 780 binary and continuous traits. I recognize that a thorough analysis including fine-mapping LD adjustment would be a lot of work, but an initial scan should definitely be feasible.
Author Response	We thank the reviewer for their excellent suggestion. To examine whether disruptive variants are more likely to be associated with GWAS phenotypes, we first parsed all GWAS SNPs found in the UK Biobank GWAS Atlas. We then calculated R^2 values between all ExAC variants in our dataset and the UK Biobank SNPs, using 1000 Genomes Phase 3 data ExAC variants in strong LD with GWAS SNPs had a disruption rate that was not significantly different from the overall disruption rate for ExAC variants as a whole (Supplementary Fig. 5a). Our results were robust to how strongly LD was defined (i.e., across multiple R^2 thresholds), and consistent regardless of if African or European alleles were used to calculate LD (Supplementary Fig. 5a, Supplementary Fig. 5c). Since most GWAS SNPs occur at $MAF \geq 0.1\%$ and a sizable fraction of our tested variants are rare, we also repeated our analysis restricted for variants at $MAF \geq 0.1\%$ but still found no significant trend (Supplementary Fig. 5b, Supplementary Fig. 5d). As a control, we also repeated these same analyses using SNPs from the NCBI GWAS Catalog and found the exact same trends as those for the UK Biobank GWAS Atlas (Supplementary Fig. 5e-h). Our analysis reveals that irrespective of whether a variant is in strong LD with a GWAS SNP or not, the likelihood that the variant is disruptive is unchanged. This potentially suggests that our pipeline is well suited for detecting phenotypic variant that may otherwise go undetected by GWAS; however, we emphasize that our analysis represents only a preliminary scan of the GWAS data available. A more thorough, rigorous analysis is required to properly dissect potential relationships between molecularly disruptive variants and GWAS SNPs. We have commented on the potential relationship between disruptive variants and GWAS SNPs in our discussion section.
Excerpt from Revised Manuscript	[Page 13] While our experimental framework was not designed to find potentially causal variants driving GWAS phenotypes (Supplementary Fig. 5; Methods), experimental frameworks that can tease apart functional variants from those that are non-functional will be key to identifying causal variants in common disease. [Methods] To examine whether ExAC variants that are in strong linkage disequilibrium (LD) with GWAS SNPs are more likely to be disruptive, we first extracted all SNPs associated with phenotypes in the UK Biobank GWAS Atlas⁸⁸. We then calculated R^2 values between all ExAC variants in our dataset and the UK Biobank GWAS SNPs, using 1000 Genomes Phase 3 data⁸⁹.

	ExAC variants in strong LD with GWAS SNPs had a disruption rate that was not significantly different from the overall disruption rate (Supplementary Fig. 5a). Results were robust across multiple R^2 thresholds using either African or European allele frequencies (Supplementary Fig. 5a, Supplementary Fig. 5c). Since most GWAS SNPs occur at $MAF \geq 0.1\%$ and a sizable fraction of our tested variants are rare, we also repeated our analysis restricted for variants at $MAF \geq 0.1\%$ but still found no significant trend (Supplementary Fig. 5b, Supplementary Fig. 5d). As a control, we also repeated these same analyses using SNPs from the NCBI GWAS Catalog⁹⁰ and found the exact same trends as those for the UK Biobank GWAS Atlas (Supplementary Fig. 5e-h).
--	---

Ref3.1 – "Several interesting observations were obtained" –

Reviewer Comment	In this manuscript, the authors describe their study on mutations disrupting protein-protein interactions (PPIs) among human genome SNVs. They extracted SNV data from ExAC dataset and performed large scale yeast two-hybrid assays to evaluate whether these mutations affect these selected PPIs. Several interesting observations were obtained: (i) PPI disruptive SNVs are prevalent (ii) Their fraction is reversely proportional to MAF (iii) They are more enriched in disease-associated genes.
Author Response	We thank the reviewer for their summary and their subsequent comments.

Ref3.2 – Biological drill-down of PPI disruption data –

Reviewer Comment	Although the biological significance and medical inference of disruptive SNVs is important, the authors do not provide any serious biological drill downs that would confirm their PPI data - they definitely need to address this issue by providing such biological drill downs on one or two selected PPI pairs. As a suggestion, the synthetic lethality, either genetically or pharmacologically, may help them to validate some of their results.
Author Response	We agree with the reviewer that the biological significance and medical inference of disruptive SNVs is important. We therefore introduce an example of two disease mutations on SMAD4 with matching disruption profiles that result in the same disease and a third non-disruptive SMAD4 mutation that results in a clinically distinct disease. We also present a physiologically-relevant case study in which we characterized a disruptive SNV in SEPT12 and further generated CRISPR-edited mice to validate the functional impact of the SNV in vivo. We begin, however, by first providing several lines of evidence to demonstrate that both the interactions and their corresponding disruptions reported in our manuscript are biologically meaningful. The ability of the yeast-two hybrid assay to detect biologically meaningful interactions across organisms has been extensively demonstrated across multiple studies (Walhout et al. Science

2000; Boulton et al. *Science* 2002; Rual et al. *Nature* 2005; Yu et al. *Science* 2008; Arabidopsis Interactome Mapping Consortium. *Science* 2011; Rolland et al. *Cell* 2014; Vo et al. *Cell* 2016). To further confirm the biological significance of the interactions used in this study, we examined the co-expression of protein abundance levels corresponding to interactions used in our study. Using protein expression levels for 30 adult and fetal tissues and cell types from the Human Proteome Map (Kim et al. *Nature* 2014), we found that proteins corresponding to interactions used in our study were significantly more likely to be co-expressed than random protein pairs, confirming the *in vivo* biological significance of the interactions used in the study (**Supplementary Note Fig. 1**). We have added this result to the revised manuscript.

To demonstrate how matching disruption profiles can provide insight towards the molecular mechanisms of disease, we highlight three mutations on SMAD4 (**Fig. 4f**), a crucial protein in the TGF β /SMAD signaling pathway. Two mutations on SMAD4, E330K and G352R, result in juvenile polyposis (Gallione et al. *Am J Med Genet* 2010; Sayed et al. *Ann Surg Oncol* 2002) while a third mutation, N13S, results in a clinically distinct disease, pulmonary arterial hypertension (Nasim et al. *Hum Mut* 2011). We observed that E330K and G352R cluster together in three dimensional space near the SMAD4-SMAD3 interaction interface (**Fig. 4g**). N13S, in contrast, is positioned away from E330K and G352R near the N-terminus of SMAD4. In agreement with the proximal clustering of E330K and G352R near the SMAD4-SMAD3 interaction interface, both mutations disrupted the SMAD4 interaction with SMAD3 in addition to disrupting the SMAD4-SMAD9 interaction (**Fig. 4f**). These SMAD protein disruption results agree with previous evidence implicating the TGF β /SMAD signaling pathway in the formation of juvenile polyposis (Jung et al. *Gastroent* 2017; Massangué. *Cell* 2008). In contrast, the N13S mutation left SMAD4 interactions with SMAD3 and SMAD9 intact, which agrees with a previous study that found no evidence that N13S alters SMAD-mediated signaling (Nasim et al. *Hum Mut* 2011). Collectively, these results demonstrate how mutations with the same disruption profiles can lead to the same disease phenotype.

As suggested by the reviewer, we also performed a physiologically-relevant study using disruptive mutations found in the GTPase, SEPT12. SEPT12 is known to interact with other septin proteins found in the SEPT2, SEPT6, and SEPT7 protein subgroups, to form a filamentous structure at the sperm annulus (Mostowy and Cossart *Nat Rev Mol Cell Biol* 2012; Sellin et al. *Mol*

	Biol Cell 2014; Kuo et al. J Cell Sci 2015). Mutation-induced perturbations to SEPT12 interactions with these septin subgroup proteins can compromise the structural integrity of the sperm annulus to result in male infertility. Indeed, an infertility-causing mutation in men, SEPT12_D197N, was previously shown to disrupt interactions with septin proteins found at the sperm annulus and was shown to result in a disorganized sperm annulus and poor sperm motility in a mouse model for SEPT12_D197N (Kuo et al. J Cell Sci 2015). We identified a rare variant in SEPT12, G169E (MAF = 0.02%), that disrupted interactions with SEPT7 as well as the SEPT2 subgroup proteins, SEPT1 and SEPT5 (Fig. 5c). The infertility-causing mutation, D197N, disrupted these exact same septin interaction partners by Y2H. Using homology modeling, we observed that both G169E and D197N mutations occur at SEPT12 interaction interface residues with SEPT1 (Fig. 5d) and confirmed by co-IP that both mutations disrupt the SEPT12-SEPT1 interaction without reducing protein stability in 293T cells (Fig. 5e). These results demonstrate that these mutations function by specifically perturbing SEPT12 protein-protein interactions as opposed to disrupting SEPT12 stability as a whole. To investigate whether these matching SEPT12 molecular phenotypes result in corresponding organismal phenotypes, we generated Sept12^{G169E} mice using a CRISPR-editing approach. We found that homozygous Sept12^{G169E} males were subfertile in comparison to wild-type males (Fig. 5f). Notably, sperm from homozygous Sept12^{G169E} males exhibited poor motility (Fig. 5g), a phenotype also reported for Sept12^{D197N} male mice (Kuo et al. J Cell Sci 2015). Taken in context with our in vitro disruption data, our mouse model strongly suggests that SEPT12_G169E is an infertility-associated mutation in men and further demonstrates how shared disruption profiles can be used to prioritize candidate disease-associated mutations. Lastly, we note that no individuals homozygous for SEPT12_G169E are reported in ExAC, potentially explaining why SEPT12_G169E has not been previously reported as an infertility-associated variant. We have updated our manuscript to include our SMAD4 and SEPT12 analyses.
Excerpt from Revised Manuscript	[Supplementary Note 1] We further note that Y2H has been extensively demonstrated to detect biologically meaningful interactions across many organisms in many studies^{1,3-8}. To further confirm the biological significance of the interactions used in this study, we examined the co-expression of protein abundance levels corresponding to interactions used in our study. Using protein expression levels

	for 30 adult and fetal tissues and cell types from the Human Proteome Map⁹, we found that proteins corresponding to interactions used in our study were significantly more likely to be co-expressed than random protein pairs, confirming the in vivo biological significance of the interactions used in the study (Supplementary Note Fig. 1). [Page 9] To demonstrate how pairs of disease-associated mutations on the same gene with matching disruption profiles can result in the same disease, we highlight three disease-associated mutations on SMAD4 (Fig. 4f), a crucial protein in the TGFβ/SMAD signaling pathway. Two mutations on SMAD4, E330K and G352R, are associated with juvenile polyposis^{54,55} while a third mutation, N13S, results in a clinically distinct disease, pulmonary arterial hypertension⁵⁶. We observed that E330K and G352R cluster together in three dimensional space near the SMAD4-SMAD3 interaction interface (Fig. 4g). N13S, in contrast, appears positioned away from E330K and G352R near the N-terminus of SMAD4. In agreement with the proximal clustering of E330K and G352R near the SMAD4-SMAD3 interaction interface, both mutations disrupted the SMAD4 interaction with SMAD3 in addition to disrupting the SMAD4-SMAD9 interaction (Fig. 4f). These SMAD protein disruption results agree with previous evidence implicating the TGFβ/SMAD signaling pathway in the formation of juvenile polyposis^{57,58}. In contrast, the N13S mutation left SMAD4 interactions with SMAD3 and SMAD9 intact, which agrees with a previous study that found no evidence that N13S alters SMAD-mediated signaling⁵⁶. [Page 10] To further show how potentially physiologically-relevant mutations can be identified using shared disruption profiles, we also characterized a pair of disruptive mutations on the GTPase, SEPT12: a rare variant not known to associate with any disease phenotypes, G169E (MAF = 0.02%), and D197N, an infertility-causing mutation in men⁶¹. Both mutations perturbed interactions with SEPT7 and SEPT2 subgroup proteins, SEPT1 and SEPT5 (Fig. 5c). These perturbations are particularly relevant because SEPT12 is known to interact with other septin proteins found in the SEPT2, SEPT6, and SEPT7 protein subgroups to form a filamentous structure at the sperm annulus⁶²⁻⁶⁴. Moreover, the infertility-causing mutation SEPT12_D197N, which was previously shown to perturb interactions with these same septin subgroup proteins, resulted in a disorganized sperm annulus and poor sperm motility in a mouse model for D197N⁶⁴. Lastly, using homology modeling, we observed that both G169E and D197N mutations occur at SEPT12 interaction interface residues with SEPT1 (Fig. 5d) and confirmed that both mutations disrupt the SEPT12-SEPT1 interaction without reducing protein stability in 293T cells (Fig. 5e). These results demonstrate that these mutations function by specifically perturbing SEPT12 protein-protein interactions as opposed to disrupting SEPT12 stability as a whole. We then investigated whether these matching SEPT12 molecular phenotypes result in corresponding organismal phenotypes by generating Sept12^{G169E} mice using a CRISPR-editing approach⁶⁵. We found that homozygous Sept12^{G169E} males were subfertile in comparison to wild-type males (Fig. 5f). Notably, sperm from homozygous Sept12^{G169E} males exhibited poor motility (Fig. 5g), a phenotype also reported for Sept12^{D197N} male mice⁶⁴. These observations of poor sperm motility and subfertility in mice suggest that SEPT12_G169E may deleteriously impact fertility in men homozygous for this mutation, although we note that no individuals homozygous for SEPT12_G169E are reported in ExAC. Taken together with our in vitro data, these results also demonstrate how shared disruption profiles can be used to prioritize candidate disease-associated mutations.
--	---

Ref3.3 – Explanation for why PSPH mutation T152I does not cause a pathological phenotype –

Reviewer Comment	In consistency with the above, the PSPH T152I SNV needs further exploration. It exhibits the same enzymatic characteristics as the disease-associated mutation D32N. Why it does not cause any pathological phenotype as D32N mutant should be explained.
Author Response	The reviewer correctly notes that despite having the same enzymatic characteristics as the disease-associated mutation D32N, the rare variant, T152I (MAF = 0.10%), does not cause any

	known pathological phenotype. We note that D32N is deleterious in a compound heterozygous background. Specifically, in a study by Veiga-da-Cunha and colleagues, the authors characterize two mutations, D32N and M52T, which are each inherited from healthy parents (Veiga-da-Cunha et al. Eur J Hum Genet 2003). In their study, D32N was reported to lower Vmax for PSPH by ~50% while M52T nearly abolishes PSPH catalytic activity. The patient therefore lacks a wild-type, functional copy of PSPH. As for T152I, we would like to point out that no T152I homozygous individuals are reported in ExAC, potentially explaining why no individuals carrying a T152I mutation have a reported pathological phenotype. We have updated our manuscript to further clarify this point.
Excerpt from Revised Manuscript	[Page 10] To do this, we tested two mutations with matching disruption profiles on the protein PSPH (Fig. 5a): (i) T152I, a rare variant (MAF = 0.10%) in ExAC that disrupts an interaction with itself and (ii) D32N, which also disrupts an interaction with itself and causes phosphoserine phosphatase deficiency in a compound heterozygous individual with two deleterious PSPH mutations⁵⁹. [Page 10] Because phosphoserine phosphatase deficiency is a recessively inherited condition⁵⁹, our findings suggest that T152I may lead to the same disease phenotype in homozygous or compound heterozygous individuals.

Ref3.4 – Distinguish deleterious effects on PPIs and on enzymatic activity –

Reviewer Comment	The authors provide two examples of functional defects of SNVs. These were demonstrated by in vitro enzymatical assays which do not require any PPI in the reactions. It is necessary to distinguish the deleterious effects on PPIs and on enzyme activities, which should not be the same.
Author Response	The reviewer comments that the deleterious effects of perturbations to protein-protein interactions should not be the same as alterations to enzymatic activity. In general, we agree with the reviewer's comment; however, PSPH is a notable exception. In our disruption assays, we observed that a PSPH rare variant, T152I, disrupted an interaction with itself while leaving all other interactions intact (Fig. 5a). Previous research has reported the importance of PSPH dimerization, noting that PSPH is a dimer in solution and that "mutant residues such as Y138K, F139K, and Y143K, which could interfere with dimer interfaces, exist in an aggregated and insoluble form" (Kim H-Y et al. JBC 2002). In this context, proper dimerization is actually crucial to PSPH enzymatic activity. Since the T152I rare variant disrupts an interaction with itself, we hypothesized that T152I would also reduce enzymatic

activity. Indeed, *in vitro* measurements of PSPH enzymatic activity showed that the T152I mutant protein reduced activity to the same extent as D32N while a non-disruptive mutation, T149M, left enzymatic activity intact (**Fig. 5b**). We further note that dimerization may also be crucial for AKR7A2 enzymatic activity since the mutation A142T both disrupts an interaction with itself (**Fig. 3f**) and reduces AKR7A2 enzymatic activity (**Fig. 3g**).

Nonetheless, changes in enzymatic activity are not evidence of particular protein-protein interactions occurring or not occurring. Moreover, disruptions to protein homodimers only comprise 61 of the 669 (9.1%) total disrupted interactions reported in our dataset. As such, in an effort to show that shared disruption profiles can be applied to other categories of interaction perturbations, we also characterized disruptive mutations on the GTPase, SEPT12.

As previously discussed in Ref3.2, we found that the SEPT12 rare variant, G169E (MAF = 0.02%), and the infertility-causing variant, D197N, each disrupted the same interactions with other septin proteins (**Fig. 5c**). We also observed that both G169E and D197N mutations occur at SEPT12 interaction interface residues with SEPT1 (**Fig. 5d**) and confirmed by co-IP that both mutations disrupt the SEPT12-SEPT1 interaction without reducing protein stability in 293T cells (**Fig. 5e**). These results demonstrate that these mutations function by specifically perturbing SEPT12 protein-protein interactions as opposed to disrupting SEPT12 stability as a whole.

Considering that the infertility-causing mutation SEPT12_D197N has been previously shown to disrupt interactions with other septin proteins and that these perturbations result in a defective sperm annulus (Kuo et al. *J Cell Sci* 2015), we then generated CRISPR-edited mice for SEPT12_G169E to investigate whether this equally disruptive variant also impacts fertility. We found that homozygous *Sept12*^{G169E} mice were subfertile (**Fig. 5f**) and that sperm from these homozygous *Sept12*^{G169E} males exhibited poor motility (**Fig. 5g**), a phenotype also reported for *Sept12*^{D197N} male mice (Kuo et al. *J Cell Sci* 2015). This result therefore confirms that SEPT12 interaction perturbations induced by the G169E rare variant are physiologically relevant and broadly demonstrates how shared disruption profiles can be used to prioritize candidate disease-associated mutations.

We have updated the manuscript to better articulate how PSPH dimerization is coupled to its enzymatic activity in addition to discussing our SEPT12 example.

Excerpt from Revised Manuscript	[Page 10] To do this, we tested two mutations with matching disruption profiles on the protein PSPH (Fig. 5a): (i) T152I, a rare variant (MAF = 0.10%) in ExAC that disrupts an interaction with itself and (ii) D32N, which also disrupts an interaction with itself and causes phosphoserine phosphatase deficiency in a compound heterozygous individual with two deleterious PSPH mutations⁵⁹. An additional PSPH non-disruptive rare variant, T149M, was included as a control. Since PSPH exists as a dimer in solution and can aggregate when mutations that interfere with dimerization are introduced⁶⁰, we reasoned that mutations that disrupt this dimerization may also reduce PSPH enzymatic activity. We therefore purified recombinant wild-type, D32N, T152I, and T149M PSPH proteins and measured for changes in phosphatase activity for PSPH mutants relative to wild-type using a malachite green assay. [Page 10] To further show how potentially physiologically-relevant mutations can be identified using shared disruption profiles, we also characterized a pair of disruptive mutations on the GTPase, SEPT12: a rare variant not known to associate with any disease phenotypes, G169E (MAF = 0.02%), and D197N, an infertility-causing mutation in men⁶¹. Both mutations perturbed interactions with SEPT7 and SEPT2 subgroup proteins, SEPT1 and SEPT5 (Fig. 5c). These perturbations are particularly relevant because SEPT12 is known to interact with other septin proteins found in the SEPT2, SEPT6, and SEPT7 protein subgroups to form a filamentous structure at the sperm annulus⁶²⁻⁶⁴. Moreover, the infertility-causing mutation SEPT12_D197N, which was previously shown to perturb interactions with these same septin subgroup proteins, resulted in a disorganized sperm annulus and poor sperm motility in a mouse model for D197N⁶⁴. Lastly, using homology modeling, we observed that both G169E and D197N mutations occur at SEPT12 interaction interface residues with SEPT1 (Fig. 5d) and confirmed that both mutations disrupt the SEPT12-SEPT1 interaction without reducing protein stability in 293T cells (Fig. 5e). These results demonstrate that these mutations function by specifically perturbing SEPT12 protein-protein interactions as opposed to disrupting SEPT12 stability as a whole. We then investigated whether these matching SEPT12 molecular phenotypes result in corresponding organismal phenotypes by generating Sept12^{G169E} mice using a CRISPR-editing approach⁶⁵. We found that homozygous Sept12^{G169E} males were subfertile in comparison to wild-type males (Fig. 5f). Notably, sperm from homozygous Sept12^{G169E} males exhibited poor motility (Fig. 5g), a phenotype also reported for Sept12^{D197N} male mice⁶⁴. These observations of poor sperm motility and subfertility in mice suggest that SEPT12_G169E may deleteriously impact fertility in men homozygous for this mutation, although we note that no individuals homozygous for SEPT12_G169E are reported in ExAC. Taken together with our in vitro data, these results also demonstrate how shared disruption profiles can be used to prioritize candidate disease-associated mutations.
---

Ref3.5 – Disruptive SNVs may be more prevalent than reported here –

Reviewer Comment	Furthermore, the estimation of many parameters was based on two PPI assays: YTH and PCA. As the data provided by the authors and work from many previous studies, the recovery rate is pretty low. This might skew the estimation of the true value of disruptive SNV fraction, and should definitely be notified.
Author Response	We agree with the reviewer that no high-throughput assay can detect all interactions (Venkatesan et al. Nature Methods 2009; Braun et al. Nature Methods 2009). If we could detect additional interactions (i.e., have an assay with a higher recovery rate), the extent of disruptions detected would only be higher. Thus, the number of disruptions that we detect, and the corresponding fraction of disruptive SNVs presented in the manuscript are lower bound estimates of the actual disruptive SNVs, further highlighting the significance of our findings.

	Additionally, detecting a lower bound for the actual fraction of disruptive SNVs does not affect any of our results on relative trends across categories (such as fraction of disruptive SNVs across allele frequencies) as we uniformly detect a lower bound on disruptive SNVs in all categories. Moreover, all the absolute values presented (such as the fraction of disruptive SNVs for common variants) would be even more significant if we could detect disruptions corresponding to all interactions. We have amended the discussion section of our manuscript to include comments about how the fraction of disruptive variants is actually a lower-bound estimate due to low recovery rate of Y2H.
Excerpt from Revised Manuscript	[Page 11] It should be noted that, like any high-throughput assay, Y2H cannot detect all interactions of a given protein. If we were able to detect more interactions, we would likely discover more interaction disruptions. Therefore, this 10.5% figure represents only a lower-bound estimate for the number of disruptive missense variants per individual.

Ref3.6 – Possibility that PPI disruptions are compensated by redundant parallel pathways –

Reviewer Comment	In addition, the authors mention that a disruption is mild and only partially interferes the involved biochemical process. I think it's also possible that the disruption can be compensated by redundant parallel pathways.
Author Response	The reviewer suggests the possibility that redundant parallel pathways may compensate for the potentially deleterious impact of disruptive SNVs. We first wish to clarify that our high-throughput assays for detecting disrupted protein-protein interactions (Y2H and PCA) are binary; each mutant protein-protein interaction pair is tested separately. As a result, redundant parallel pathways cannot compensate for any PPI disruptions observed in our binary interaction assays. Nonetheless, the reviewer is correct to point out that within a cellular context, redundant parallel pathways can compensate for the otherwise potentially deleterious impact of disrupted PPIs. Motivated by the reviewer's excellent point, we explored the extent to which redundant parallel pathways may actually offset the potentially deleterious impact of disruptive variants. While several anecdotal examples of redundant parallel pathways have been described in literature, we note that redundant, duplicated genes are yet to be defined more systematically. Since duplicate genes can also compensate for corresponding proteins impacted by a disruptive mutation, we took advantage of the Duplicated Genes Database which lists 3,543 duplicate genes across 945 different gene groups (Ouedraogo et al. PLOS ONE 2012). We then

	examined whether disruptive SNVs occurred more frequently within genes with reported duplications in comparison to non-disruptive SNVs but found no significant differences between both groups (~12% and ~10%, respectively; P value = 0.55; Supplementary Fig. 3a). To validate the robustness of this result, we further defined our own set of functionally similar proteins by performing BLAST for the human proteome against itself and generating a list of protein groups in which constitutive proteins share greater than 90% sequence identity and coverage. Again, we found that the fraction of SNVs that are disruptive were not enriched within duplicate gene groups in comparison to non-disruptive SNVs (Supplementary Fig. 3b). Furthermore, this result held across all sequence identity and coverage cutoff values (Supplementary Fig. 3b). These results therefore indicate that while at the cellular level disruptive variants can be compensated by genes with redundant functions, such compensation does not appear to affect PPI disruption rates. We have amended our manuscript to include these analyses.
Excerpt from Revised Manuscript	[Page 8] Notably, duplicate or functionally similar genes can compensate for corresponding proteins impacted by a disruptive mutation. However, we found no enrichment for disruptive variants within a published set of duplicate genes⁵¹ in comparison to non-disruptive variants (Supplementary Fig. 3a), nor within a custom-generated set of sequence-conserved, functionally similar proteins (Supplementary Fig. 3b; Methods). [Methods] Duplicate genes were obtained from the Duplicated Genes Database⁵¹ which lists 3,543 duplicate genes across 945 different gene groups. In order to compare the robustness of these duplicate gene definitions across many different cutoffs, we additionally defined our own metric for protein similarity by running a BLAST of the human proteome against itself and eliminating all pairs of proteins with less than 40% sequence identity. The remaining pairs were scored using a weighted combination of the pair's percent identity and the coverage with respect to each protein. In Supplementary Fig. 3b, we flexibly defined duplicate genes as all pairs of genes whose score meets a minimal duplication threshold tested across all valid ranges (where 0 for "Duplication Threshold" represents no appreciable similarity and 1 represents perfect identity). Score is calculated as: $Score = \alpha * PercentIdentity * Coverage_{Avg} + (1 - \alpha) * Coverage_{Avg}$ where $\alpha = 0.95$ and $Coverage_{Avg}$ is the average coverage between both proteins.

Ref3.7 – Better discussion for medical inference of disruptive SNVs –

Reviewer Comment	Moreover, the authors did not describe the whole scenario of the medical inference. They suggested that a SNV might contribute liability to certain genetic diseases when combined with mutations in other genes involved in the same biochemical process. However, this is a rare situation.
Author Response	For Mendelian recessive disease mutations, the reviewer correctly comments that a single mutation, present in a homozygote, is

sufficient for causing disease. Indeed, we and others have systematically explored the impact of Mendelian disease mutations on protein-protein interactions (Wei et al. *PLOS Genet* 2014; Sahni et al. *Cell* 2015). But many Mendelian diseases also feature high levels of heterogeneity, where there are multiple distinct variants that can cause disease. In these cases, individuals with Mendelian disease are instead heterozygous for two distinct, deleterious mutations (a separate mutation on each gene copy). The compound heterozygous D32N mutation on PSPH is one such example in which the D32N mutation in conjunction with a second deleterious mutation, M52T, results in disease. It is now commonly accepted that for many Mendelian disorders, most cases are in fact heterozygous for two distinct mutant alleles.

In the case of complex disease, including cancer and heart disease however, a single mutated gene is not the cause of the disease, but rather multiple mutations on more than one gene are needed to cause the disorder. Each disease-associated variant in complex disease therefore contributes a certain measure of disease risk that can be quantified by a GWAS, and some authors consider these effects to be approximately additive across loci (Visscher and Goddard. *Genetics* 2019). It is also possible, as exemplified in our study, to functionally assess the effects of mutations on protein function through experimental assays.

Studying complex disease is difficult since very often no single variant alone is fully penetrant. Nonetheless, the simplest case of complex disease, digenic inheritance in which two genes both contribute to a single phenotype, is actually quite prevalent. Searching HGMD (Stenson et al. *Human Genet* 2017) yields a total of 365 mutations that contribute to digenic inheritance. Moreover, a search through PubMed for “digenic mutations” yielded 378 papers, although this strictly refers to cases in which two heterozygous mutations in different genes must occur together for the disease phenotype to manifest. Cases in which the impact of a disease-causing mutation in one gene is influenced by a polymorphic variant in another gene are far more common and are extensively documented in HGMD. Such variants are often only partially penetrant, resulting in disease in only particular genetic backgrounds (Cooper et al. *Hum Genet* 2013). While dissecting how these variants modulate each other’s impact is not straightforward, individually assessing the impact of these variants in isolation is a crucial first step towards understanding how these variants function epistatically. In this context, our study represents an important resource for examining what fraction of population

	variants are functional and could conceivably play a role in disease risk as a result. We have added this discussion to our revised manuscript.
Excerpt from Revised Manuscript	[Page 11] Co-occurrence of AKR7A2_A142T with similarly disruptive mutations in ABAT or SSADH could therefore result in a neurological disorder that would not otherwise occur in an individual harboring only a single disruptive mutation. Such relationships are frequent in complex disease, including cancer and heart disease, which unlike Mendelian mutations, require multiple mutations on more than one gene to cause a disorder. Each disease-associated mutation in complex disease therefore contributes a certain measure of disease risk that can be quantified by a GWAS, and some authors consider these effects to be approximately additive across loci⁷². Measuring how one mutation modulates the impact of another is challenging; however, measuring which mutations are individually functional is a crucial first step. Hence, we anticipate that our SNV-interaction network will serve as a pivotal framework for defining the epistatic relationships that modulate the impact of disruptive variants, particularly for partially penetrant variants that only result in disease in certain genetic backgrounds (Supplementary Note 4). [Supplementary Note 4] Studying complex disease is difficult since very often no single variant alone is fully penetrant. Nonetheless, the simplest case of complex disease, digenic inheritance in which two genes both contribute to a single phenotype, is actually quite prevalent. Searching HGMD¹⁰ yields a total of 365 mutations that contribute to digenic inheritance. Moreover, a search through PubMed for “digenic mutations” yielded 378 papers, although this strictly refers to cases in which two heterozygous mutations in different genes must occur together for the disease phenotype to manifest. Cases in which the impact of a disease-causing mutation in one gene is influenced by a polymorphic variant in another gene are far more common and are extensively documented in HGMD. Such variants are often only partially penetrant, resulting in disease in only particular genetic backgrounds¹¹. While dissecting how these variants modulate each other’s impact is not straightforward, individually assessing the impact of these variants in isolation is a crucial first step towards understanding how these variants function epistatically. In this context, our study represents an important resource for examining what fraction of population variants are functional and could conceivably play a role in disease risk as a result.

Ref3.8 – Pharmacogenomic and toxicogenomic implications of our disruptive SNV data –

Reviewer Comment	I believe the results if this study might have more implications in pharmacogenomics and toxicogenomics. For example, the disruptive SNVs may affect the sensitivities to certain drugs and environmental cues.
Author Response	We thank the reviewer for their excellent suggestion. We absolutely agree that the results in this study carry implications for pharmacogenomics and toxicogenomics. Disruptive SNVs on enzymes may alter the metabolic kinetics of impacted enzymes, while SNVs on transporters and targets of drugs may lead to changes in the pharmacokinetic and pharmacodynamic properties of their corresponding proteins. These effects combined can contribute to variability of drug efficacy and toxicity among the human population. For example, the D816H/V mutations on a receptor tyrosine kinase, KIT, confers resistance to imatinib and sunitinib by shifting the conformational equilibrium of KIT (Gajiwala et al. PNAS 2009). The S9G population variant on dopamine

	receptor 3 (DRD3) is associated with a lower risk of gastrointestinal side effects in response to levodopa treatment (Rieck et al. Pharmacogenomics J. 2018). In our study, we also characterized the impact of a common variant A142T (MAF = 6.4%) on the enzyme AKR7A2 and showed that A142T significantly reduced the specific activity of AKR7A2 for its native substrate, succinic semialdehyde. Notably, AKR7A2 is an aldo-keto reductase that can metabolize anti-cancer drugs, including doxorubicin and daunorubicin. Indeed, a previous study characterizing the effects of common SNVs on aldo-keto reductases found that AKR7A2_A142T significantly decreased the in vitro metabolism of both doxorubicin and daunorubicin by AKR7A2, which could have important implications in cancer therapy (Bains et al. J. Pharmacol. Exp. Ther. 2010). This agreement between the disruption profile of AKR7A2_A142T and its compromised enzymatic activity for both the native and drug substrates of AKR7A2 suggests that disruptive SNVs on enzymes may be excellent candidates for identifying mutations that compromise drug-protein interactions. To further demonstrate the potential application of our dataset to pharmacogenomics and toxicogenomics, we intersected our dataset with four sets of genes: all human enzymes, drug-metabolizing enzymes, drug targets, and drug transporters. The list of all human enzyme genes was obtained from HumanCyc version 21.5 (Romero, et al. Genome Biol 2004), while the lists of drug-related genes were obtained from DrugBank version 5.1.2 (Wishart et al. Nucleic Acids Res 2017). Among the SNVs that we tested, 350 were on enzymes, and 84 of them disrupted at least one interaction. More specifically, 54 SNVs were tested on drug-metabolizing enzymes and 12 of them were disruptive. In addition, 227 SNVs were tested on drug targets, 66 of which disrupted at least one interaction. Lastly, five SNVs were tested on drug transporters and three of them were disruptive. These numbers highlight the potential impact of SNVs on variability of drug response and toxicity among the population. We have added a table (Supplementary Table 9) summarizing these numbers and have included these points in the Discussion section.
Excerpt from Revised Manuscript	[Page 7] In addition to impacting SSA turnover, the A142T mutation is reported to significantly decrease the in vitro metabolism of both doxorubicin and daunorubicin by AKR7A2, which could have important implications in cancer therapy⁴⁷. [Page 12]

	The results of our study may carry important implications in related fields such as pharmacogenomics and toxicogenomics. Disruptive SNVs on enzymes may alter the metabolic kinetics of impacted enzymes, while SNVs on transporters and targets of drugs may lead to changes in the pharmacokinetic and pharmacodynamic properties of their corresponding proteins. For example, the D816H/V mutations on the receptor tyrosine kinase, KIT, confers resistance to imatinib and sunitinib by shifting the conformational equilibrium of KIT⁷³. As a potential resource to pharmacogenomics and toxicogenomics, we provide a table of all disruptive SNVs potentially relevant to drug action (Supplementary Table 9; Methods). [Methods] To generate a dataset of disruptive SNVs potentially relevant to pharmacogenomics and toxicogenomics, we intersected our dataset with four sets of genes: all human enzymes, drug-metabolizing enzymes, drug targets, and drug transporters. The list of all human enzyme genes was obtained from HumanCyc version 21.5⁹¹, while the lists of drug-related genes were obtained from DrugBank version 5.1.2⁹². Among the SNVs that we tested, 350 were on enzymes, and 84 of them disrupted at least one interaction. More specifically, 54 SNVs were tested on drug-metabolizing enzymes and 12 of them were disruptive. In addition, 227 SNVs were tested on drug targets, 66 of which disrupted at least one interaction. Lastly, five SNVs were tested on drug transporters and three of them were disruptive. These numbers highlight the potential impact of SNVs on variability of drug response and toxicity among the population. These SNVs are provided in Supplementary Table 9.
--	---

Reviewers' Comments:

Reviewer #1:

Remarks to the Author:

The manuscript has been reworked and improved substantially, including expansion on examples and more in death discussion. Since I was excited about the paper already in the first round, I recommend acceptance as is.

Reviewer #2:

Remarks to the Author:

Thanks for your responses to my comments. I am a little disappointed (!) that there is no enrichment for disruption of GWAS-linked coding variants, but the result is clean and I think adds to the manuscript.

Reviewer #3:

Remarks to the Author:

The authors successfully addressed all of my questions and I therefore recommend this manuscript for publication.